# Robust Distortion-Free Watermark for Autoregressive Audio Generation Models

**Yihan Wu**[*], **Georgios Milis**[*], **Ruibo Chen**[*], **Heng Huang**[†]
Department of Computer Science
University of Maryland, College Park
{ywu42, milis, rbchen, heng}@umd.edu

## Abstract

The rapid advancement of next-token-prediction models has led to widespread adoption across modalities, enabling the creation of realistic synthetic media. In the audio domain, while autoregressive speech models have propelled conversational interactions forward, the potential for misuse, such as impersonation in phishing schemes or crafting misleading speech recordings, has also increased. Security measures such as watermarking have thus become essential to ensuring the authenticity of digital media. Traditional statistical watermarking methods used for autoregressive language models face challenges when applied to autoregressive audio models, due to the inevitable "retokenization mismatch" - the discrepancy between original and retokenized discrete audio token sequences. To address this, we introduce ALIGNED-IS, a novel, distortion-free watermark, specifically crafted for audio generation models. This technique utilizes a clustering approach that treats tokens within the same cluster equivalently, effectively countering the retokenization mismatch issue. Our comprehensive testing on prevalent audio generation platforms demonstrates that ALIGNED-IS not only preserves the quality of generated audio but also significantly improves the watermark detectability compared to the state-of-the-art distortion-free watermarking adaptations, establishing a new benchmark in secure audio technology applications. We release the code in https://github.com/g-milis/AlignedIS.

## 1 Introduction

Autoregressive audio generation models (Lakhotia et al., 2021; Rubenstein et al., 2023; Borsos et al., 2022; Zhang et al., 2023; Nguyen et al., 2025; Zhan et al., 2024; Ge et al., 2023; Lu et al., 2024), such as those enabling sophisticated voice synthesis, have significantly advanced in mimicking human-like speech. As these models become integral to various applications, from virtual assistants to real-time translation, their potential misuse also escalates. Malicious uses include impersonating individuals in phishing attacks, fabricating audio for misinformation, and automating scam calls with natural-sounding voices. Additionally, the spread of synthetic audio can undermine the authenticity of digital communication and pose challenges in legal contexts where recording verification is crucial. To address these concerns, implementing robust and detectable watermarks in synthetic audio becomes essential, ensuring traceability and accountability in the use of generative models while safeguarding against their unauthorized exploitation.

Statistical watermarking techniques are a promising method to identify machine-generated content from autoregressive language models (Kirchenbauer et al., 2023; Liu & Bu, 2024; Chen et al., 2025a).

---

[*]Equal contribution
[†]This work was partially supported by NSF IIS 2347592, 2348169, DBI 2405416, CCF 2348306, CNS 2347617, RISE 2536663.

39th Conference on Neural Information Processing Systems (NeurIPS 2025).

However, due to *retokenization mismatch* (Wu et al., 2025), directly applying them to audio generation models leads to poor detectability. Unlike autoregressive language models, where text tokenization is deterministic and reversible, autoregressive audio models incorporate an additional encoder and decoder that map from audio to discrete tokens and back. In the process of audio generation, an audio prompt is encoded into a sequence of tokens. Subsequently, this sequence undergoes next-token prediction to generate an output sequence, which is then decoded back into audio form. Watermark detection requires re-encoding the generated audio into its tokenized form. However, this retokenized token sequence does not exactly match the original in-generation sequence, a phenomenon we refer to as *retokenization mismatch*.

To tackle this challenge, we introduce ALIGNED-IS, a novel, robust, and distortion-free watermark specifically designed for audio generation models. We observe that mismatch arises from different discretization of similar continuous features, which should be close in the encoder and decoder's feature spaces. Leveraging this insight, we developed a clustering-based watermarking framework that considers tokens within the same cluster as equivalent. We summarize our contributions as follows:

- We develop ALIGNED-IS, the first distortion-free watermark for autoregressive audio generation models. We identify the retokenization mismatch phenomenon and we propose a novel clustering-based distortion-free watermarking algorithms to address this challenge.
- Through comprehensive experiments, we validate the distortion-freeness, detectability, and robustness of ALIGNED-IS on popular open-source audio generation models. Our results show a significantly improvement in detectability compared to directly applying existing distortion-free watermarks to audio generation models.

## 2    Related Work

**Audio generation models.**    With the success of large language models, researchers have developed multimodal foundation models that extend transformers to handle continuous signals through modality-specific encoders and decoders (Vaswani et al., 2017). Lakhotia et al. (2021) first reframed pure audio generation as a language-modeling task (Lakhotia et al., 2021), which gave rise to AudioLM (Borsos et al., 2022) and later AudioPaLM's integration of text generation capabilities (Rubenstein et al., 2023). Building on this, SpeechGPT (Zhang et al., 2023) and SpiritLM (Nguyen et al., 2025) discretize HuBERT features (Hsu et al., 2021) into semantic units that a fine-tuned LLaMA (Touvron et al., 2023) consumes as an expanded vocabulary of text and audio tokens. The same discretization strategy supports fine-grained tasks such as voice cloning from text and a discretized voice prompt (Chen et al., 2024b). Other models-including SEED-LLaMA (Ge et al., 2023), Unified-IO 2 (Lu et al., 2024), and AnyGPT (Zhan et al., 2024)-represent audio and images with discrete token sequences alongside text. In contrast, CoDi-2 (Tang et al., 2024), VITA (Fu et al., 2024), and NExT-GPT (Wu et al., 2024) employ a transformer decoder that directly processes continuous feature vectors together with text token embeddings, treating multimodal outputs as regression while still using next-token prediction for text.

**Post hoc audio watermarking.**    Embedding watermarks into host audio dates back decades (Lie & Chang, 2006), leveraging human insensitivity to mid-frequency bands. More recently, deep-learning methods use autoencoder architectures to invisibly encode payloads into a frequency transform of the signal (Liu et al., 2024a; Chen et al., 2023). The work of San Roman et al. (2024) further introduces temporally localized watermarking with time-dependent detection, and error-correcting codes have been employed to boost robustness (Wu et al., 2023a). However, these approaches degrade output quality and offer no formal statistical guarantee for reliable watermark detection. Furthermore, they are fragile to waveform perturbations, and require an additional processing step to embed them.

**Distortion-free watermark.**    Aaronson (2022) introduced a pioneering distortion-free watermarking approach that utilizes Gumbel-Softmax to alter token distributions. Christ et al. (2023) and Kuditipudi et al. (2023) applied inverse-sampling and Gumbel-Softmax, respectively, to modify the token distributions in watermarked content, employing watermark keys based on either token positioning or predetermined key lists. However, the technique from Christ et al. (2023) exhibits limited resilience when altered and lacks empirical evidence of its detectability. In contrast, Kuditipudi et al. (2023)'s method demands extensive resampling from the secret key distribution for detection, proving

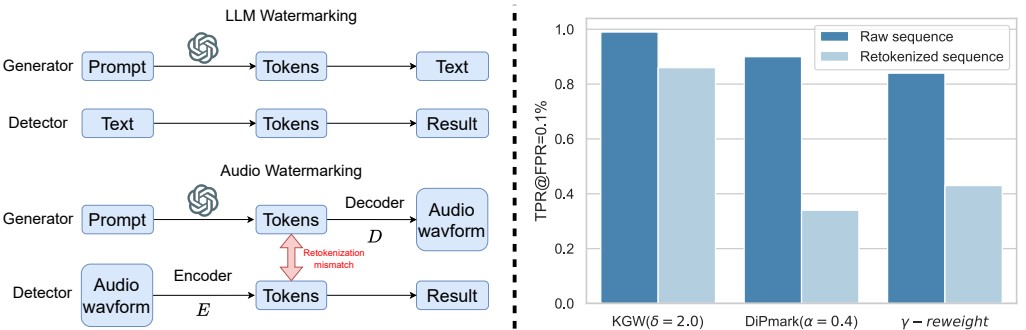

Figure 1: **Left:** Illustration of the retokenization mismatch problem during audio watermark detection. **Right:** Performance comparison of audio watermark detection on raw versus retokenized audio sequences. We evaluate KGW watermark (Kirchenbauer et al., 2023) DiPmark (Wu et al., 2023b), and $\gamma$-reweight (Hu et al., 2023), reporting the true positive rate at a fixed 1% false positive rate.

inefficient for extensive texts. Hu et al. (2023) proposed inverse-sampling and reweight strategies for watermarking, though their detection method is not model-agnostic and requires access to the language model API and specific prompts. Wu et al. (2023b) refined the reweight technique and introduced a model-agnostic detection mechanism. Dathathri et al. (2024) proposed SynthID, which enables distortion-freeness of LM watermarking with multiple generations.

## 3 Preliminary

**Notations.** We follow the notations used in (Hu et al., 2023). The vocabulary (or token) set is denoted by $V$ and its cardinality by $N = |V|$. We define the set $\mathcal{V}$, which includes all possible token sequences including those of zero length and the set $\mathcal{A}$, which includes all possible audios. Within an autoregressive audio generation model, a token sequence is generated based on a specific prompt. At any given step, the probability of producing the next token $x_{n+1} \in V$, given the preceding sequence $x_1, \ldots, x_n$, is denoted by $P_M(x_{n+1} \mid x_1, x_2, \ldots, x_n)$. For simplicity and clarity, we adopt a more concise notation: $P_M(\boldsymbol{x}_{n+1:n+m} \mid \boldsymbol{x}_{1:n})$, where $\boldsymbol{x}_{n+1:n+m} = (x_{n+1}, \ldots, x_{n+m})$. It is important to note that the prompt is intentionally excluded from this notation. In audio generation models, we denote the audio-token encoder by $E(\cdot) : \mathcal{A} \to \mathcal{V}$ and the token-audio decoder, or vocoder, by $D(\cdot) : \mathcal{V} \to \mathcal{A}$.

### 3.1 Statistical Watermarks

In watermarking applications, the service provider employs a set of *i.i.d.* watermark codes $\{\theta_i \in \Theta, i \in \mathbb{N}\}$, defined over the code space $\Theta$. Each code $\theta_i$ is typically derived from a secret key $\mathsf{key} \in \mathcal{K}$ and the n-gram preceding context, denoted $\boldsymbol{x}_{t-n:t-1}$.

In the watermark generator, a reweight strategy is used to embed a statistical signal into the generated content. Let $\mathcal{P}$ denote the set of all probability distributions over the token set $V$. The reweight strategy is a function $P_W : \mathcal{P} \times \Theta \to \mathcal{P}$. For the token distribution at the $(n+1)$-th generation step, $P_M(x_{n+1} \mid \boldsymbol{x}_{1:n}) \in \mathcal{P}$, the watermarked distribution is defined by $P_W(P_M(x_{n+1} \mid \boldsymbol{x}_{1:n}), \theta_i)$. For brevity, this is represented as $P_W(x_{n+1} \mid \boldsymbol{x}_{1:n}, \theta_i)$. A distortion-free watermark ensures that the averaged distribution $P_W(x_{n+1} \mid \boldsymbol{x}_{1:n}, \theta_i)$ with respect to $\theta_i$ is equal to the original distribution $P_M(x_{n+1} \mid \boldsymbol{x}_{1:n})$.

**Definition 3.1** (Distortion-free watermark). Given the watermark code set $\Theta$, a distribution $\mathcal{P}_\Theta$ on $\Theta$, original LM distribution $P_M$, and the watermarked distribution $P_W(\cdot | \theta \in \Theta)$ A distortion-free watermark should satisfy $\forall x \in V$,

$$\mathbb{E}_{\theta \sim P_\Theta}\left[P_W(x \mid \boldsymbol{x}_{1:n}, \theta)\right] = P_M(x \mid \boldsymbol{x}_{1:n}).$$

Current popular distortion-free strategies include Gumbel-softmax (Aaronson, 2022), inverse-sampling (Christ et al., 2023; Hu et al., 2023; Kuditipudi et al., 2023) and reweight-based strategy (Wu et al., 2023b; Dathathri et al., 2024; Feng et al., 2025; Chen et al., 2025a).

During watermark detection, the user only has access to the watermark key, the reweight strategy, and the generated audio. The detector employs a hypothesis testing approach to ascertain the presence of the watermark signal. The null hypothesis $H_0$ is defined as *"The content is generated without the presence of watermarks"*. The detector adopts a score function based on the watermark key and the reweight strategy, which exhibits statistical bias between the watermarked and unwatermarked token sequences.

## 3.2 Autoregressive audio generation models

Autoregressive audio models typically begin by encoding raw waveforms into self-supervised feature representations, such as those produced by wav2vec (Baevski et al., 2020) or HuBERT (Hsu et al., 2021). These continuous features are then quantized into discrete semantic units. In contrast to text-where tokens generally correspond to reversible character or subword units (Sennrich et al., 2016)-an audio waveform is segmented into overlapping frames and processed by a feature extractor. To discretize the feature space, a clustering algorithm (e.g., $k$-means) partitions it into $N_u$ clusters, with each frame assigned to the nearest cluster centroid. The resulting sequence of centroid indices serves as "audio tokens" for the language model. During inference, these discrete tokens are transformed back into a waveform using a neural vocoder (Kong et al., 2020). In multimodal text–audio LLMs, the overall vocabulary size is $N = N_u + N_t$, where $N_t$ denotes the number of text tokens (Zhang et al., 2023; Nguyen et al., 2025).

**Retokenization mismatch.**    Let $x$ denote the token sequence output by the audio generation model, where the final audio is obtained by decoding through the token-audio decoder $D(x)$, referred to as vocoder. For watermark detection, the generated audio is passed through the audio-token encoder $E(D(x))$, and a hypothesis test is conducted on $E(D(x))$ to identify the presence of a watermark signal. Nonetheless, $E(D(x))$ often differs from the original $x$, which weakens the statistical evidence captured by the detection scores. This discrepancy can be viewed as an unavoidable token-level perturbation occurring during the detection process. As shown in Figure 1, there is a large gap between the detectability of the raw token sequence $x$ and the retokenized sequence $E(D(x))$.

# 4 Methodology

To address the retokenization mismatch issue in autoregressive audio generation models, we developed ALIGNED-IS, a clustering-based watermarking framework. In ALIGNED-IS, audio tokens are initially segmented into clusters. Subsequently, we introduce a novel, distortion-free reweight strategy, aligned inverse sampling, tailored specifically for these clustered tokens. During watermark generation, this reweight strategy alters the output distribution according to the clusters. For watermark detection, the detector verifies whether the current token corresponds to the expected cluster, as determined by the reweight strategy and the embedded watermark code. Since mismatched tokens are likely within the same cluster, the detector can still accurately capture the correct statistical signal, even in the presence of retokenization mismatches.

In this section, we first present the clustering method for tackling retokenization mismatch and a distortion-free reweighting strategy, aligned inverse sampling, which is specifically devised for the clustered tokens. Then, we introduce our general watermark algorithm and detection statistic.

## 4.1 Audio token clustering

During clustering, the objective is to split all audio tokens into distinct clusters based on their similarity. It is crucial to note that the clustering algorithm is executed only once for each audio generation model, its results are stored, and can be accessed during watermark generation and detection. Consequently, this approach does not increase the computational cost during sampling. To achieve this, we collect the audio token embeddings $\{e_1, ..., e_m\}$ based on the token-audio encoder $E$ and use $k$-means algorithm to generate the corresponding clusters $\{c_1, ..., c_h\}$. We employ $k$-means since the discretization process uses the Euclidean norm of audio feature vectors to discretize them, by selecting the nearest token embedding vector. While acessing the embedding vectors in open-source models is trivial, a closed source model provider can still implement our watermarking method and expose its detection API.

Table 1: The effect of clustering on retokenization mismatch across datasets for SpiritLM.

| Dataset | Mismatch Rate Before | Mismatch Rate After | Reduction (%) |
|---|---|---|---|
| MMW Book Report | 0.3749 | 0.2117 | 43.55% |
| MMW Story | 0.3652 | 0.2174 | 40.47% |
| MMW Fake News | 0.4295 | 0.2300 | 46.44% |
| Dolly CW | 0.3634 | 0.2134 | 41.30% |
| Longform QA | 0.3757 | 0.2109 | 43.85% |
| Finance QA | 0.3587 | 0.2133 | 40.54% |

With our segmentation strategy, mismatched tokens are more likely to be in the same cluster because they are supposed to share similar embeddings in the encoder $E$. The watermark generator and detector can utilize the cluster information to avoid the detectability reduction caused by the retokenization mismatch. The inverse sampling watermark is a distortion-free method that can be applied directly to clustering scenarios.

To quantify the effectiveness of our clustering method in mitigating retokenization mismatch, we report the token mismatch rates before and after applying clustering on SpiritLM across multiple datasets in Table 1. The mismatch rate is computed by comparing the original response tokens with those obtained after a decode-then-encode retokenization process. As shown in the table, our clustering method significantly reduces the mismatch rate, demonstrating its effectiveness in aligning tokens during watermarking generation and detection.

We have also evaluated other popular clustering methods, e.g., (Gaussian mixture models, spectral clustering) and found that they produce similar results. We did not explore methods beyond clustering, as it is the most natural and direct approach to address the retokenization mismatch. Empirically, clustering has proven effective in significantly reducing such mismatches.

## 4.2 Aligned inverse sampling

**Inverse Sampling.** Let $c_1, \ldots, c_h$ represent the identified clusters, and let $\Pr(c_1), \ldots, \Pr(c_h)$ denote the sum of the token probabilities within them, such that $\Pr(c_i) = \sum_{x \in c_i} P_M(x)$. It is natural to map these probabilities onto the interval [0,1] and employ inverse sampling to pseudo-randomly select a number $r(\theta) \in [0, 1]$ seeded by the watermark code $\theta$. The subsequent token is then sampled from cluster $c_i$ if $r$ falls within the interval $[\sum_{j=0}^{i-1} \Pr(c_j), \sum_{j=0}^{i} \Pr(c_j)]$, with $\Pr(c_0) := 0$. During detection, cluster information and the pseudo-random number $r(\theta)$ can be reconstructed using the watermark code $\theta$, allowing us to compute a statistical score by comparing the cluster against $r(\theta)$.

However, since token probabilities are unknown during watermark detection, we cannot ascertain the cluster probabilities $\Pr(c_1), \ldots, \Pr(c_h)$. This discrepancy leads to an alignment issue between $r(\theta)$ and the identified cluster $c_i$, as it becomes uncertain whether $r(\theta)$ is within the interval $[\sum_{j=0}^{i-1} \Pr(c_j), \sum_{j=0}^{i} \Pr(c_j)]$. To address this, Christ et al. (2023); Kuditipudi et al. (2023) proposed a position-based statistical score for watermark detection. The underlying principle is that if $r(\theta)$ is close to 1, then the index of the selected cluster during watermark generation is likely close to $h$ (the end of the clusters). However, the detection methods proposed in Kuditipudi et al. (2023) cannot provide a theoretical guarantee of the false positive rate. Besides, this position-based score has shown low detectability, as evidenced in Kuditipudi et al. (2023) and our experimental findings. To address the flaws of inverse sampling, we introduce aligned inverse sampling, which substantially improves detection efficiency while providing a provable guarantee on the false-positive rate.

**Aligned Inverse Sampling.** Consider the scenario where $\Pr(c_1) = \cdots = \Pr(c_h) = \frac{1}{h}$. In this case, the pseudo-random number $r$ and the identified cluster $c_i$ are aligned. Specifically, if $r(\theta) \in \left[\frac{i-1}{h}, \frac{i}{h}\right]$, the detector can confidently assert that $c_i$ was generated through inverse sampling with $r(\theta)$. A natural enhancement to inverse sampling involves rearranging the cluster probabilities within the interval [0,1] to emulate this aligned scenario.

Details on this method are in Algorithm 3.

**Theorem 4.1.** *Aligned inverse sampling is a distortion-free watermark.*

---

**Algorithm 1** ALIGNED-IS generator.

---

1: **Input:** secret key key, prompt $\boldsymbol{x}_{-m:0}$, generate length $t \in \mathbb{N}$, token-audio decoder $D$.
2: Initialize watermark code history $hist$.
3: **for** $i = 1, \ldots, t$ **do**
4:     Calculate the token distribution for generating the $i$-th token $P_M(\cdot \mid \boldsymbol{x}_{-m:i-1})$.
5:     Generate a watermark code $\theta_i = (\text{key}, \boldsymbol{x}_{i-n,i-1})$.
6:     **if** $\theta_i \in hist$ **then**
7:         Sample the next token $x_i$ using original distribution $P_M(\cdot|\boldsymbol{x}_{-m:i-1})$
8:     **else**
9:         Generate the pseudo-random number $r(\theta_i)$.
10:        Calculate watermarked distribution $P_W(\cdot|\boldsymbol{x}_{-m:i-1})$ via aligned inverse sampling.
11:        Sample the next token $x_i$ using distribution $P_W(\cdot|\boldsymbol{x}_{-m:i-1})$.
12: **return** audio waveform $D(\boldsymbol{x}_{1:t})$.

---

---

**Algorithm 2** ALIGNED-IS detector.

---

1: **Input:** audio waveform $a$, audio-token encoder $E$, secret key key, score function $s$, threshold $z$.
2: calculate token sequence $\boldsymbol{x}_{1:t} = E(a)$
3: Initialize the score function: $S = 0$.
4: **for** $i = 2, ..., t$ **do**
5:     Generate the watermark code $k_i = (\text{key}, \boldsymbol{x}_{i-n,i-1})$.
6:     Generate the pseudo-random number $r(\theta_i)$.
7:     Update the score function via $S = S + s(r(\theta_i), x_i)$.
8: **return** $S > z$.

---

*Proof.* The proof is straightforward. As aligned inverse sampling does not modify the cluster probability, by the property of inverse sampling, we have $P_M(x) = \mathbb{E}_\theta[P_W(x|\theta)]$. □

With the aforementioned probability assignments, we can define a statistical score for the detector based on the pseudo-random number $r$ and the current token $x$.

**Definition 4.2.** Given the pseudo-random number $r$ and the current token $x$, the detection score $s(r, x)$ is defined as

$$s(r, x) := \begin{cases} 1 & \text{if } r \in \left[\frac{i-1}{h}, \frac{i}{h}\right], \text{ where } x \in c_i, \\ 0 & \text{otherwise.} \end{cases} \tag{1}$$

Note, when $\Pr(c_i) < \frac{1}{h}$, aligned inverse sampling may sample tokens from clusters other than $c_i$ even if the pseudo-random number $r$ falls within $\left[\frac{i-1}{h}, \frac{i}{h}\right]$. This can reduce detection accuracy. However, empirical results indicate that the overall detectability of aligned inverse sampling still surpasses that of the regular inverse sampling reweight strategy.

### 4.3    ALIGNED-IS

With the aligned inverse sampling reweight strategy, we can construct our watermarking algorithm ALIGNED-IS. ALIGNED-IS consists of a watermark generator and a watermark detector. In the watermark generator, the watermarked token at step $t$ is generated using a secret key key and the prior n-gram content $\boldsymbol{x}_{t-n,t-1}$ as the watermark code $\theta_t$. The pseudo-random number $r_t(\theta_t)$ is then generated based on $\theta_t$. Following the inverse sampling reweight strategy, the cluster $c_{i(\theta_t)}$ is selected based on $r_t(\theta_t)$. The next token $x_t$ is sampled by randomly selecting a token within $c_{i(\theta_t)}$ according to the token probability of the original language model $P_M(\cdot|\boldsymbol{x}_{1:t-1})$. Following Hu et al. (2023), we use a watermark code history $hist$ to ensure the distortion-freeness for multiple generation. If the current watermark code is in $hist$, we will sample from the original model's distribution instead of the watermarked distribution. The algorithm is detailed in Alg. 1.

During detection, access to the generated audio and the watermark key key is required. The token sequence $\boldsymbol{x}_{1:t}$ is first recovered using the encoder $E$ of the audio generation model. Then, for each $i = 1, \ldots, t$, the watermark code $\theta_i$ is generated based on the watermark key and the n-gram

Table 2: Detectability comparison of watermarking methods on MMW Book Report, MMW Story, and MMW Fake News with SpiritLM. We report true positive rate at 1% and 0.1% false positive rate and the median p-value.

| Method | MMW Book Report | | | MMW Story | | | MMW Fake News | | |
|---|---|---|---|---|---|---|---|---|---|
| | TPR@FPR | | Median $p$-value | TPR@FPR | | Median $p$-value | TPR@FPR | | Median $p$-value |
| | 1% | 0.1% | | 1% | 0.1% | | 1% | 0.1% | |
| KGW($\delta$=1.0) | 0.75 | 0.40 | 0.002 | 0.66 | 0.42 | 0.004 | 0.63 | 0.38 | 0.003 |
| KGW($\delta$=1.5) | 0.93 | 0.76 | 1.1e-05 | 0.96 | 0.90 | 1.2e-05 | 0.87 | 0.71 | 1.3e-04 |
| KGW($\delta$=2.0) | 0.97 | 0.94 | 2.8e-07 | 0.97 | 0.89 | 5.1e-07 | 0.92 | 0.86 | 2.5e-06 |
| Unigram($\delta$=1.0) | 0.09 | 0.00 | 0.241 | 0.06 | 0.02 | 0.241 | 0.08 | 0.01 | 0.198 |
| Unigram($\delta$=1.5) | 0.27 | 0.12 | 0.043 | 0.28 | 0.11 | 0.042 | 0.37 | 0.20 | 0.040 |
| Unigram($\delta$=2.0) | 0.56 | 0.29 | 0.006 | 0.55 | 0.30 | 0.005 | 0.53 | 0.28 | 0.007 |
| $\gamma$-reweight | 0.68 | 0.41 | 0.003 | 0.53 | 0.30 | 0.006 | 0.68 | 0.43 | 0.002 |
| DiPmark($\alpha$=0.3) | 0.55 | 0.27 | 0.009 | 0.35 | 0.21 | 0.029 | 0.44 | 0.26 | 0.017 |
| DiPmark($\alpha$=0.4) | 0.63 | 0.41 | 0.002 | 0.57 | 0.33 | 0.004 | 0.61 | 0.34 | 0.003 |
| ITS | 0.77 | 0.64 | 2.0e-04 | 0.82 | 0.70 | 2.0e-04 | 0.83 | 0.74 | 2.0e-04 |
| ALIGNED-IS | 0.88 | 0.77 | 8.0e-05 | 0.92 | 0.80 | 4.6e-05 | 0.97 | 0.82 | 4.9e-05 |

Table 3: Detectability comparison of watermarking methods on Dolly CW, Longform QA, and Librispeech with SpiritLM. We report true positive rate at 1% and 0.1% false positive rate and the median p-value.

| Method | Dolly CW | | | Longform QA | | | Librispeech | | |
|---|---|---|---|---|---|---|---|---|---|
| | TPR@FPR | | Median $p$-value | TPR@FPR | | Median $p$-value | TPR@FPR | | Median $p$-value |
| | 1% | 0.1% | | 1% | 0.1% | | 1% | 0.1% | |
| KGW($\delta$=1.0) | 0.69 | 0.44 | 0.003 | 0.75 | 0.44 | 0.001 | 0.69 | 0.42 | 0.002 |
| KGW($\delta$=1.5) | 0.93 | 0.83 | 2.4e-05 | 0.95 | 0.84 | 1.8e-05 | 0.97 | 0.86 | 1.3e-05 |
| KGW($\delta$=2.0) | 0.98 | 0.90 | 6.0e-07 | 0.99 | 0.92 | 3.4e-07 | 0.99 | 0.97 | 9.7e-08 |
| Unigram($\delta$=1.0) | 0.10 | 0.02 | 0.268 | 0.15 | 0.06 | 0.164 | 0.06 | 0.01 | 0.268 |
| Unigram($\delta$=1.5) | 0.32 | 0.16 | 0.057 | 0.40 | 0.17 | 0.021 | 0.25 | 0.07 | 0.060 |
| Unigram($\delta$=2.0) | 0.43 | 0.23 | 0.016 | 0.56 | 0.32 | 0.006 | 0.48 | 0.23 | 0.012 |
| $\gamma$-reweight | 0.63 | 0.43 | 0.002 | 0.56 | 0.35 | 0.005 | 0.71 | 0.46 | 0.001 |
| DiPmark($\alpha$=0.3) | 0.45 | 0.19 | 0.019 | 0.50 | 0.25 | 0.010 | 0.60 | 0.31 | 0.006 |
| DiPmark($\alpha$=0.4) | 0.58 | 0.34 | 0.005 | 0.53 | 0.29 | 0.009 | 0.69 | 0.45 | 0.002 |
| ITS | 0.79 | 0.62 | 4.0e-04 | 0.86 | 0.73 | 2.0e-04 | 0.90 | 0.83 | 2.0e-04 |
| ALIGNED-IS | 0.92 | 0.80 | 2.3e-05 | 0.96 | 0.82 | 1.6e-05 | 0.95 | 0.84 | 6.7e-06 |

content. Subsequently, the pseudo-random number $r_i(\theta_i)$ is recovered based on $\theta_i$. Next, we calculate the statistical score $s(r_i(\theta_i), x_i)$ following Definition 4.2. The final statistic is given by $S(\boldsymbol{x}_{1:t}) = \sum_{i=1}^{t} s(r_i(\theta_i), x_i)$.

Under the null hypothesis, $S(\boldsymbol{x}_{1:t})$ follows a binomial distribution with a success rate of $\frac{1}{h}$. Thus, we have the following tail bound derived from the Hoeffding's inequality: $\Pr(S(\boldsymbol{x}_{1:t}) \geq k) \leq \exp(-2t(\frac{1}{h} - \frac{k}{t})^2)$. By setting a threshold on the false positive rate (e.g. FPR=1%), we can calculate a threshold $z$ by solving $\exp(-2t(\frac{1}{h} - \frac{z}{t})^2) = 0.01$. If the score $S(\boldsymbol{x}_{1:t})$ is greater than $z$, we reject the null hypothesis and claim that the sentence is watermarked. The detection algorithm is in Alg. 2.

## 5 Experiments

**Baselines.** We evaluate the performance of our methods against various statistical watermarking baselines, including two biased watermarking approaches, KGW (Kirchenbauer et al., 2023) and Unigram (Zhao et al., 2023), as well as three unbiased watermarking algorithms, $\gamma$-reweight (Hu et al., 2023), DiPmark (Wu et al., 2023b), and ITS-edit (Kuditipudi et al., 2023). We also compare our in-generation method with state-of-the-art post hoc watermarking methods in Appendix C.

**Models and Datasets.** We evaluate our watermarking approach ALIGNED-ISon raw audios generated by text-speech aligned conversational models. This allows us for flexibility to prompt speech generation with both text and speech prompts. We use the models SpiritLM (Nguyen et al., 2025)

Table 4: Detectability comparison of watermarking methods on Dolly CW, Longform QA, and Finance QA with SpeechGPT. We report true positive rate at 1% and 0.1% false positive rate and the median p-value.

| | Dolly CW | | | Longform QA | | | Finance QA | | |
|---|---|---|---|---|---|---|---|---|---|
| | TPR@FPR | | Median $p$-value | TPR@FPR | | Median $p$-value | TPR@FPR | | Median $p$-value |
| Method | 1% | 0.1% | | 1% | 0.1% | | 1% | 0.1% | |
| KGW($\delta$=1.0) | 0.32 | 0.14 | 0.048 | 0.40 | 0.19 | 0.023 | 0.36 | 0.21 | 0.024 |
| KGW($\delta$=1.5) | 0.56 | 0.36 | 0.005 | 0.72 | 0.48 | 0.001 | 0.68 | 0.52 | 8.2e-04 |
| KGW($\delta$=2.0) | 0.73 | 0.51 | 7.4e-04 | 0.84 | 0.69 | 4.2e-05 | 0.80 | 0.70 | 7.5e-05 |
| Unigram($\delta$=1.0) | 0.17 | 0.06 | 0.122 | 0.25 | 0.10 | 0.054 | 0.21 | 0.07 | 0.061 |
| Unigram($\delta$=1.5) | 0.40 | 0.21 | 0.023 | 0.55 | 0.26 | 0.006 | 0.64 | 0.37 | 0.003 |
| Unigram($\delta$=2.0) | 0.57 | 0.38 | 0.004 | 0.71 | 0.54 | 6.1e-04 | 0.72 | 0.59 | 3.0e-04 |
| $\gamma$-reweight | 0.37 | 0.15 | 0.047 | 0.46 | 0.24 | 0.015 | 0.56 | 0.32 | 0.007 |
| DiPmark($\alpha$=0.3) | 0.23 | 0.11 | 0.089 | 0.37 | 0.12 | 0.027 | 0.33 | 0.17 | 0.032 |
| DiPmark($\alpha$=0.4) | 0.29 | 0.12 | 0.049 | 0.46 | 0.24 | 0.014 | 0.54 | 0.25 | 0.008 |
| ALIGNED-IS | 0.82 | 0.69 | 5.6e-05 | 0.95 | 0.90 | 1.8e-0 | 0.94 | 0.89 | 1.7e-07 |

Table 5: Robustness comparison of watermarking methods for Longform QA with SpiritLM under signal processing attacks. We report TPR at 1% FPR. 'Dist.' refers distorted watermarks and 'Dist. free' refers distortion-free watermarks.

| | Watermark | No attack | Echo (0.05sec) | Gauss. noise (30dB) | Lowpass (40%) | Smooth (6 samp.) | Speed (0.9) | Speed (1.1) |
|---|---|---|---|---|---|---|---|---|
| Dist. | KGW($\delta$=1.0) | 0.75 | 0.64 | 0.77 | 0.77 | 0.80 | 0.13 | 0.11 |
| | KGW($\delta$=1.5) | 0.95 | 0.95 | 0.97 | 0.96 | 0.96 | 0.24 | 0.27 |
| | KGW($\delta$=2.0) | 0.99 | 0.97 | 0.99 | 0.99 | 0.99 | 0.35 | 0.35 |
| | Unigram($\delta$=1.0) | 0.15 | 0.15 | 0.03 | 0.16 | 0.11 | 0.01 | 0.02 |
| | Unigram($\delta$=1.5) | 0.40 | 0.41 | 0.13 | 0.39 | 0.30 | 0.07 | 0.05 |
| | Unigram($\delta$=2.0) | 0.56 | 0.57 | 0.30 | 0.56 | 0.47 | 0.13 | 0.08 |
| Dist. free | $\gamma$-reweight | 0.56 | 0.56 | 0.44 | 0.58 | 0.56 | 0.17 | 0.09 |
| | DiP($\alpha$=0.3) | 0.50 | 0.50 | 0.35 | 0.48 | 0.49 | 0.09 | 0.10 |
| | DiP($\alpha$=0.4) | 0.53 | 0.52 | 0.39 | 0.54 | 0.49 | 0.14 | 0.10 |
| | ITS | 0.86 | 0.85 | 0.79 | 0.88 | 0.85 | 0.06 | 0.00 |
| | ALIGNED-IS | 0.96 | 0.93 | 0.93 | 0.94 | 0.94 | 0.31 | 0.28 |

and SpeechGPT (Zhang et al., 2023) for the speech generation tasks. For text prompting, we follow Kirchenbauer et al. (2023); Hu et al. (2023) and include three MMW datasets (Piet et al., 2023), Dolly CW (Conover et al., 2023), and two tasks from WaterBench (Tu et al., 2023). For speech prompting, we use the validation set of LibriSpeech (Panayotov et al., 2015).

**Watermarking parameters.** We evaluate the detectability of ALIGNED-IS on the speech generation task with different audio generation models. We generate 500 examples for each task. We use the prefix 1-gram together with a secret key as the watermark keys. We select $\alpha \in \{0.3, 0.4\}$ for DiPmark, and $\delta \in \{1.0, 1.5, 2.0\}$ and $\gamma = 0.5$ for KGW watermark (Kirchenbauer et al., 2023), $\delta \in \{1.0, 1.5, 2.0\}$ for Unigram (Zhao et al., 2023). For ALIGNED-IS, we partition the token-embedding space into 20 clusters using the $k$-means algorithm, then perform linear sum assignment to ensure that the resulting centroids are sufficiently separated to accommodate potential retokenization errors. We justify the choice of $h = 20$ in Appendix E.1. All experiments are conducted on a NVIDIA A6000 GPU.

## 5.1 Detectability

Following the evaluation metric of the previous works (Kirchenbauer et al., 2023; Wu et al., 2023b), we report the true positive rate at guaranteed false positive rates, i.e., TPR@FPR=$\{1\%, 0.1\%\}$. Notice, as the detectors of ITS-edit do not provide a theoretical guarantee, we report the true positive rate at the estimated false positive rate following their detecting algorithms.

From Table 2, 3, and 4 we see that ALIGNED-IS achieved the best detectability comparing with all other unbiased watermarks, at least 10% improvement on all TPR@FPR metrics. Besides, ALIGNED-

Table 6: Robustness comparison of watermarking methods for Longform QA with SpiritLM under codec-based, quantizing, and denoising attacks. We report TPR at 1% FPR. 'Dist.' refers distorted watermarks and 'Dist. free' refers distortion-free watermarks.

| | Watermark | EnCodec (24kHz) | MP3 (32kbps) | MP3 (40kbps) | Opus (16kbps) | Opus (31kbps) | Quant. (64-bit) | Denoise |
|---|---|---|---|---|---|---|---|---|
| Dist. | KGW($\delta$=1.0) | 0.64 | 0.51 | 0.53 | 0.65 | 0.74 | 0.68 | 0.65 |
| | KGW($\delta$=1.5) | 0.93 | 0.88 | 0.86 | 0.92 | 0.95 | 0.92 | 0.93 |
| | KGW($\delta$=2.0) | 0.98 | 0.94 | 0.93 | 0.97 | 0.98 | 0.97 | 0.97 |
| | Uni($\delta$=1.0) | 0.06 | 0.12 | 0.12 | 0.12 | 0.14 | 0.00 | 0.06 |
| | Uni($\delta$=1.5) | 0.26 | 0.32 | 0.34 | 0.33 | 0.37 | 0.10 | 0.22 |
| | Uni($\delta$=2.0) | 0.41 | 0.53 | 0.49 | 0.49 | 0.53 | 0.20 | 0.38 |
| Dist. free | $\gamma$-reweight | 0.51 | 0.42 | 0.46 | 0.54 | 0.56 | 0.57 | 0.52 |
| | DiP($\alpha$=0.3) | 0.46 | 0.30 | 0.34 | 0.46 | 0.48 | 0.60 | 0.44 |
| | DiP($\alpha$=0.4) | 0.49 | 0.41 | 0.41 | 0.47 | 0.51 | 0.56 | 0.48 |
| | ITS | 0.77 | 0.74 | 0.78 | 0.80 | 0.88 | 0.63 | 0.71 |
| | ALIGNED-IS | 0.92 | 0.80 | 0.84 | 0.91 | 0.92 | 0.88 | 0.92 |

IS outperformed the biased watermarking algorithm KGW and Unigram in most cases, and achieved comparable performance with the strongly biased KGW($\delta$=2.0).

**Time Efficiency.** Similar to KGW, Unigram, and DiPmark watermarking approaches, the minimal computational overhead introduced by the ALIGNED-IS generator occurs solely during the adjustment of token probabilities in the generation stage, and can be elegantly impelemented as a logits processor in popular deep learning framweorks. Moreover, the ALIGNED-IS detector is model-agnostic.

## 5.2 Robustness

Audio in the wild is subject to various modifications. Hosting platforms use codecs for efficiency, and users employ editing software either for recreational purposes, or specifically to erase watermarking signals. To evaluate the robustness of ALIGNED-IS under realistic channel conditions, we apply a diverse suite of **thirteen** single-channel audio attacks. We follow the no-box attacks of AudioMarkBench (Liu et al., 2024b), which includes common signal porcessing modifications (time dilation, echo, Gaussian noise addition, lowpass filtering, temporal smoothing, quantization), popular audio codings (Opus, EnCodec (Défossez et al., 2023), and MP3), as well as the denoising attack of López-López et al. (2024). The detailed settings are in Section D.1.

The results are summarized in Table 5 and Table 6. We report the true positive rate at a fixed false positive rate of 1% for each watermark across a range of attack types and strengths. ALIGNED-IS consistently exhibits the strongest robustness, outperforming all distortion-free watermarking baselines in reliably detecting watermarked audio under adversarial conditions. Experiments on two additional datasets per model are included in F.2.

## 5.3 Audio quality

We employ non-intrusive speech quality metrics to validate unbiasedness, due to the inherent stochasticity of token-based statistical watermarking methods. Specifically, we utilize the Fréchet Audio Distance (FAD) (Kilgour et al., 2019), a metric designed to directly quantify the divergence between two distributions. We compute the FAD scores between the watermarked audios and their unwatermarked counterparts generated from identical prompts and the unwatermarked model. As a baseline (no watermarking scenario), we generate two distinct sets of audio samples from the unwatermarked model, using the same prompts but different random seeds, and calculate the FAD scores between these two sets. Figure 2 shows that ALIGNED-IS and the other distortion-free watermarking schemes preserve audio quality on par with the unwatermarked baseline, whereas distortion-based approaches such as KGW and Unigram noticeably degrade it.

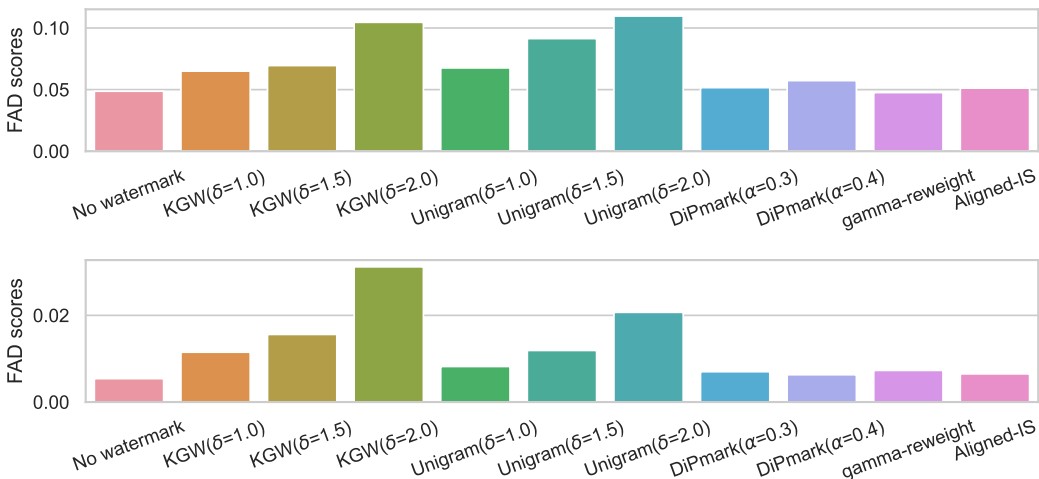

Figure 2: Audio-quality impact of watermarking methods. **Top:** FAD scores on the Dolly CW dataset evaluated with SpiritLM. **Bottom:** FAD scores on the LibriSpeech dataset evaluated with SpiritLM.

## 6  Conclusion

In conclusion, we propose ALIGNED-IS, a novel distortion-free watermarking framework tailored specifically for autoregressive audio generation models. Leveraging aligned inverse sampling, ALIGNED-IS ensures traceability and accountability in synthetic audio outputs without any degradation in audio quality. Comprehensive empirical evaluations across diverse datasets and audio generation architectures demonstrate the efficacy and robustness of our approach. Our method thus represents a meaningful advancement in watermarking technology, enhancing the security and integrity of synthetic audio and supporting trustworthy digital communication.

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

# A  Limitations

ALIGNED-IS method relies on the assumption that the retokenization mismatch is adequately captured by clustering. However, there is no formal guarantee that the clustering process fully captures all forms of retokenization errors, especially when new audio patterns or novel speech artifacts are introduced. Besides, for each new model, we need to perform the clustering of tokens, which introduces an additional computational step.

# B  Missing Algorithms

---

**Algorithm 3** Aligned inverse sampling.

---

1: **Input:** Cluster probabilities $\Pr(c_1), ..., \Pr(c_h)$, sorted from max to min probabilities. Watermark code $\theta$.
2: Initialize an overlapped_dict to store the overlapped regions.
3: Initialize a cluster_list and a prob_list for inverse sampling.
4: # *Rearrange probabilities of clusters within [0,1].*
5: **for** $i = 1, \ldots, h$ **do**
6:     cluster_list.append($c_i$)
7:     **if** $\Pr(c_i) \geq 1/h$ **then**
8:         prob_list.append($1/h$)
9:         # *Store the overlapped regions*
10:         overlapped_dict.add($\{c_i : \Pr(c_i) - 1/h\}$)
11:     **else**
12:         prob_list.append($\Pr(c_i)$)
13:         diff=$1/h - \Pr(c_i)$
14:         # *Use the overlapped regions to fill the empty region*
15:         **while** diff$> 0$ **do**
16:             **for** $j \in$ overlapped_dict **do**
17:                 cluster_list.append(j)
18:                 prob_list.append($\min\{$diff, overlapped_dict[j]$)$
19:                 diff=diff-overlapped_dict[j]
20:                 overlapped_dict[j]=$\max\{0,$ overlapped_dict[j] - diff$\}$
21: # *Sampling from the rearranged interval.*
22: Pseudo-randomly sampling $r(\theta) \in [0, 1]$ seeded by $\theta$.
23: Find $i$ s.t. $r(\theta) \in [\text{sum(prob\_list[0: i])}, \text{sum(prob\_list[0: i+1])})$
24: Randomly sample the token $x$ (following their original probability) from cluster_list[i]
25: **return** $x$

---

# C  Comparison with post hoc methods

We compare our in-generation method with the state-of-the-art post hoc watermarking methods AudioSeal (San Roman et al., 2024) and WavMark (Chen et al., 2023). Our findings indicate that, as established in existing literature, post hoc watermarking methods are neither robust, nor distortion-free. Statistical watermarking is a promising solution for embedding zero-bit watermarks to distinguish artificially generated audio. Specifically, our audio-aware method achieves a new state-of-the-art in robustness and distortion-freeness.

## C.1  Detectability & Robustness

First, we verify the results of O'Reilly et al. (2025), who identified that post hoc methods are robust to a few attacks, but extremely vulnerable to others, making them unpractical in real-world applications. Token-based statistical watermarking is more robust to a wide variety of attacks, since the watermarking information is embedded into the generated audio itself, in cotrast to overlaying mid-frequency content on the audio. We present our results in Tables 7 through 10 for SpiritLM, and Tables 11 through 14 for SpeechGPT. Notice that post hoc methods are mostly robust to low-frequency

Table 7: Robustness comparison of post hoc watermarking methods for Longform QA with SpiritLM under signal processing attacks.

| Watermark | No attack | Echo (0.05sec) | Gauss. noise (30dB) | Lowpass (40%) | Smooth (6 samp.) | Speed (0.9) | Speed (1.1) |
|---|---|---|---|---|---|---|---|
| AudioSeal | 1.00 | 0.88 | 0.07 | 1.00 | 1.00 | 0.00 | 0.00 |
| WavMark | 1.00 | 1.00 | 1.00 | 1.00 | 1.00 | 1.00 | 1.00 |
| ALIGNED-IS | 0.96 | 0.93 | 0.93 | 0.94 | 0.94 | 0.31 | 0.28 |

Table 8: Robustness comparison of post hoc watermarking methods for Longform QA with SpiritLM under codec-based and quantizing attacks.

| Watermark | EnCodec (24kHz) | MP3 (32kbps) | MP3 (40kbps) | Opus (16kbps) | Opus (31kbps) | Quant. (64-bit) |
|---|---|---|---|---|---|---|
| AudioSeal | 0.00 | 0.18 | 0.81 | 0.00 | 0.90 | 0.00 |
| WavMark | 0.00 | 1.00 | 1.00 | 0.96 | 1.00 | 0.06 |
| ALIGNED-IS | 0.92 | 0.80 | 0.84 | 0.91 | 0.92 | 0.88 |

perturbations (since they alter inaudible frequencies) and higher-bitrate codings, but are very fragile otherwise.

Detectability is reported as the TPR at 1% FPR for ALIGNED-IS, however the post hoc methods do not offer a theoretical detectability guarantee. For AudioSeal, the detection result is the probability of the audio being watermarked, so we report that probability thresholded at 99%. For WavMark, a 16-bit payload is embedded into the audio, and detection returns either an empty, or a decoded payload. We consider successful detection when a payload is not empty, even if it has a non-zero bit error rate.

## C.2 Audio quality

We then showcase that post hoc methods are inevitably harmful to audio quality, since operating on the waveform level significantly deteriorates the FAD score. Notably, audios processed by WavMark have audible high-frequency artifacts, which justifies its higher detectability. We present the audio quality in Tables 15 through 18. The mean opinion score (MOS) is also reported using the estimators NISQA and DNSMOSPro, from Mittag et al. (2021) and Cumlin et al. (2024).

## D Experimental settings

### D.1 Attack Suite

To evaluate the robustness of ALIGNED-IS under realistic channel conditions, we apply a suite of **thirteen** single-channel audio attacks. Each attack is tuned to a moderate, perceptually acceptable strength.

- **Echo (0.05 s)** - Echo with a 50 ms delay.

Table 9: Robustness comparison of post hoc watermarking methods for LibriSpeech with SpiritLM under signal processing attacks.

| Watermark | No attack | Echo (0.05sec) | Gauss. noise (30dB) | Lowpass (40%) | Smooth (6 samp.) | Speed (0.9) | Speed (1.1) |
|---|---|---|---|---|---|---|---|
| AudioSeal | 1.0 | 0.82 | 0.05 | 1.00 | 1.00 | 0.00 | 0.00 |
| WavMark | 1.00 | 1.00 | 1.00 | 1.00 | 1.00 | 1.00 | 1.00 |
| ALIGNED-IS | 0.95 | 0.92 | 0.90 | 0.93 | 0.92 | 0.28 | 0.21 |

Table 10: Robustness comparison of post hoc watermarking methods for LibriSpeech with SpiritLM under codec-based and quantizing attacks.

| Watermark | EnCodec (24kHz) | MP3 (32kbps) | MP3 (40kbps) | Opus (16kbps) | Opus (31kbps) | Quant. (64-bit) |
|---|---|---|---|---|---|---|
| AudioSeal | 0.00 | 0.18 | 0.84 | 0.00 | 0.89 | 0.00 |
| WavMark | 0.00 | 1.00 | 1.00 | 0.99 | 1.00 | 0.06 |
| ALIGNED-IS | 0.90 | 0.75 | 0.80 | 0.89 | 0.91 | 0.85 |

Table 11: Robustness comparison of post hoc watermarking methods for Longform QA with SpeechGPT under signal processing attacks.

| Watermark | No attack | Echo (0.05sec) | Gauss. noise (30dB) | Lowpass (40%) | Smooth (6 samp.) | Speed (0.9) | Speed (1.1) |
|---|---|---|---|---|---|---|---|
| AudioSeal | 1.00 | 0.77 | 0.00 | 1.00 | 1.00 | 0.00 | 0.00 |
| WavMark | 1.00 | 1.00 | 1.00 | 1.00 | 1.00 | 1.00 | 1.00 |
| ALIGNED-IS | 0.95 | 0.93 | 0.93 | 0.95 | 0.92 | 0.38 | 0.22 |

Table 12: Robustness comparison of post hoc watermarking methods for Longform QA with SpeechGPT under codec-based and quantizing attacks.

| Watermark | EnCodec (24kHz) | MP3 (32kbps) | MP3 (40kbps) | Opus (16kbps) | Opus (31kbps) | Quant. (64-bit) |
|---|---|---|---|---|---|---|
| AudioSeal | 0.00 | 0.02 | 0.05 | 0.00 | 0.26 | 0.00 |
| WavMark | 0.00 | 1.00 | 1.00 | 0.74 | 1.00 | 0.30 |
| ALIGNED-IS | 0.91 | 0.82 | 0.84 | 0.95 | 0.96 | 0.85 |

Table 13: Robustness comparison of post hoc watermarking methods for Finance QA with SpeechGPT under signal processing attacks.

| Watermark | No attack | Echo (0.05sec) | Gauss. noise (30dB) | Lowpass (40%) | Smooth (6 samp.) | Speed (0.9) | Speed (1.1) |
|---|---|---|---|---|---|---|---|
| AudioSeal | 1.00 | 0.84 | 0.01 | 1.00 | 1.00 | 0.00 | 0.00 |
| WavMark | 1.00 | 1.00 | 0.99 | 1.00 | 1.00 | 1.00 | 1.00 |
| ALIGNED-IS | 0.94 | 0.94 | 0.94 | 0.94 | 0.93 | 0.38 | 0.24 |

Table 14: Robustness comparison of post hoc watermarking methods for Finance QA with SpeechGPT under codec-based, quantizing, and denoising attacks.

| Watermark | EnCodec (24kHz) | MP3 (32kbps) | MP3 (40kbps) | Opus (16kbps) | Opus (31kbps) | Quant. (64-bit) |
|---|---|---|---|---|---|---|
| AudioSeal | 0.00 | 0.01 | 0.06 | 0.00 | 0.28 | 0.01 |
| WavMark | 0.00 | 1.00 | 1.00 | 0.74 | 1.00 | 0.29 |
| ALIGNED-IS | 0.91 | 0.89 | 0.88 | 0.95 | 0.94 | 0.87 |

Table 15: Quality comparison with post hoc methods for Dolly CW with SpiritLM

| Method | FAD ↓ | MOS ↑ | |
|---|---|---|---|
| | | NISQA | DNSMOSPro |
| No watermark | 0.0487 | 3.915 | 3.713 |
| AudioSeal | 0.3083 | 3.813 | 3.779 |
| WavMark | 1.8233 | 4.167 | 4.215 |
| ALIGNED-IS | 0.0512 | 3.860 | 3.763 |

Table 16: Quality comparison with post hoc methods for Longform QA with SpiritLM

| Method | FAD ↓ | MOS ↑ | |
| --- | --- | --- | --- |
| | | NISQA | DNSMOSPro |
| No watermark | 0.0337 | 3.934 | 3.766 |
| AudioSeal | 0.3061 | 3.830 | 3.811 |
| WavMark | 1.7910 | 4.160 | 4.190 |
| ALIGNED-IS | 0.0416 | 3.958 | 3.764 |

Table 17: Quality comparison with post hoc methods for Finance QA with SpiritLM

| Method | FAD ↓ | MOS ↑ | |
| --- | --- | --- | --- |
| | | NISQA | DNSMOSPro |
| No watermark | 0.0233 | 3.964 | 3.768 |
| AudioSeal | 0.3081 | 3.901 | 3.777 |
| WavMark | 1.7228 | 4.212 | 4.158 |
| ALIGNED-IS | 0.0515 | 3.948 | 3.766 |

- **Gaussian noise (30 dB SNR)** - Injects additive white Gaussian noise to achieve an output signal-to-noise ratio of 30 dB.

- **Low-pass filter (40 % of Nyquist)** - Applies a low-pass filter with a cut-off frequency equal to 40% of the Nyquist rate.

- **Smoothing (6-sample moving average)** - Applies a moving-average filter of width 6 samples.

- **Speed perturbation (0.9× and 1.1×)** Interpolates the waveform to speed up or slow down, accordingly.

- **EnCodec (24 kHz)** - Re-encodes the audio with Meta's EnCodec neural codec (Défossez et al., 2023) at 24 kHz bandwidth.

- **MP3 recompression (32 kbit/s and 40 kbit/s)** - Re-encodes the waveform using at the given constant bit rate.

- **Opus recompression (16 kbit/s and 31 kbit/s)** - Re-encodes the waveform at the given constant bit rate.

- **Quantization (64 levels)** - Uniformly quantizes samples to 64 discrete amplitude levels.

- **Denoising** - Applies the DCCRN (Hu et al., 2020) denoising network to the waveforms perturbed by the Gaussian noise at 30 dB.

Each attack is applied independently to the model outputs and no attack stacking is used.

Table 18: Quality comparison with post hoc methods for LibriSpeech with SpiritLM

| Method | FAD ↓ | MOS ↑ | |
| --- | --- | --- | --- |
| | | NISQA | DNSMOSPro |
| No watermark | 0.0054 | 3.966 | 3.780 |
| AudioSeal | 0.2918 | 3.915 | 3.798 |
| WavMark | 1.6509 | 4.243 | 4.203 |
| ALIGNED-IS | 0.0065 | 3.959 | 3.798 |

Table 19: Ablation on the optimal number of clusters for SpiritLM.

| h | TPR@FPR=1% | Median p-value |
|---|---|---|
| 10 | 0.798343 | 0.000473744 |
| 20 | 0.922018 | 2.27798e-05 |
| 30 | 0.724234 | 0.00165162 |
| 40 | 0.904494 | 6.96696e-05 |
| 80 | 0.696884 | 0.00124669 |
| 100 | 0.858696 | 0.000264207 |
| 120 | 0.676093 | 0.00145675 |
| 150 | 0.687927 | 0.002948 |

Table 20: Ablation on the optimal number of clusters for SpeechGPT.

| h | TPR@FPR=1% | Median p-value |
|---|---|---|
| 10 | 0.666667 | 0.000672034 |
| 20 | 0.816092 | 5.56062e-05 |
| 30 | 0.660377 | 0.00143836 |
| 40 | 0.548023 | 0.00592188 |
| 80 | 0.754144 | 0.000392686 |
| 100 | 0.248538 | 0.0748774 |
| 120 | 0.507553 | 0.0092863 |
| 150 | 0.784703 | 0.000100627 |

## E  Additional Ablations.

### E.1  Number of clusters

We conducted an ablation study on the number of clusters for both models using the Dolly CW dataset, with generation settings identical to the main experiments. We resent the results in Tables 19 and 20, which show that detectability initially increases with more clusters but begins to decline beyond a certain point. This trend arises because SpiritLM has approximately 500 audio tokens, and using too many clusters leads to overly fine partitions that fail to effectively mitigate retokenization errors. We find that strikes a good balance and yields optimal detectability, supporting our choice in the main experiments. For the non-clustering baseline, we assign tokens randomly to h=20 clusters and apply ALIGNED-IS.

### E.2  Dependence on generation length

The length of audio token sequences used for detection is typically around 500, which corresponds to approximately 10–20 seconds of audio. The duration is affected by the tokenizer and vocoder frame rates. Some variation arises because the duration of each token may differ depending on the model's duration prediction. In Table 21, we present detection results on randomly cropped segments of watermarked audio with varying durations. As shown, detection performance improves with longer audio segments, achieving reliable detectability for audio durations exceeding 5 seconds.

### E.3  Robustness to time shift

Due to the reliance on accurate audio retokenization which assumes frame alignment, the proposed watermark is not inherently robust to misalignment attacks, like speed modifications, as established in Tables 5 and 6. However, cropping attacks or time shifts can deceive detection only if they operate on small offsets (smaller than the frame size). We performed an experiment with the Longform QA dataset and the SpiritLM model, whose frame size is 645 samples. We uniformly sampled 8 time shifts, with shifts larger than 322 being equivalent to negative shifts of less than 50% of the frame length. In Table 22, we observe our watermark's strength decrease and then increase, but still maintaining high detectability at small offsets. An effective practical defense against cropping attacks would be to simply run detection on a few slightly misaligned versions of the subject audio.

Table 21: The impact of audio length on detection, for the Longform QA dataset with SpeechGPT.

| Time (sec) | TPR@FPR=1% | Median p-value |
|---|---|---|
| ∼2.0 | 0.5000 | 1.0306e-02 |
| ∼3.0 | 0.6707 | 1.7270e-03 |
| ∼4.0 | 0.8232 | 4.8318e-04 |
| ∼5.0 | 0.8659 | 7.5983e-05 |
| ∼6.0 | 0.9207 | 2.1367e-05 |
| ∼7.0 | 0.9329 | 8.7637e-06 |
| ∼8.0 | 0.9390 | 4.2917e-06 |
| ∼9.0 | 0.9451 | 2.0837e-06 |

Table 22: The impact of audio length on detection, for the Longform QA dataset with SpeechGPT.

| offset (samples) | TPR@FPR=1% | Median p-value |
|---|---|---|
| 0 | 0.94 | 2.8014e-05 |
| 80 | 0.92 | 1.2973e-05 |
| 160 | 0.76 | 0.0001 |
| 240 | 0.58 | 0.0041 |
| 320 | 0.43 | 0.0191 |
| 400 | 0.37 | 0.0318 |
| 480 | 0.54 | 0.0069 |
| 560 | 0.88 | 0.0001 |
| 640 | 0.94 | 2.6169e-05 |

### E.4 Quality comparison

We evaluate speaker similarity using WavLM-Base (Chen et al., 2022), and ASR-CER/WER using HuBERT-large fine-tuned model (Hsu et al., 2021), with watermarked audios. We use SpiritLM with Librispeech dataset. From the results in Table 23, we observe that distortion-free watermarking methods, e.g., Aligned-IS, DiPmark, and $\gamma$-reweight, achieve performance comparable to the unwatermarked baseline across all metrics. In contrast, biased and post-hoc watermarking methods noticeably degrade the generation quality.

## F   Supplementary Experimental results.

### F.1 Detectability

We present an additional watermark detectability evaluation on the Finance QA dataset with SpiritLM in Table 24. From Table 24 we see that ALIGNED-IS achieved the best detectability comparing with all other unbiased watermarks, at least 10% improvement on all TPR@FPR metrics. Besides,

Table 23: Comparison of watermarking methods on speech quality and recognition metrics. Higher Speaker Similarity and lower ASR-CER/WER indicate better performance.

| Method | Speaker Similarity ↑ | ASR-CER ↓ | ASR-WER ↓ |
|---|---|---|---|
| Baseline | 0.7550 | 0.1039 | 0.1475 |
| Audioseal | 0.7479 | 0.1086 | 0.1648 |
| Wavmark | 0.7311 | 0.1143 | 0.1819 |
| Aligned-IS | 0.7548 | 0.1038 | 0.1472 |
| DiPmark(0.4) | 0.7543 | 0.1045 | 0.1468 |
| $\gamma$-reweight | 0.7546 | 0.1043 | 0.1477 |
| KGW(1.5) | 0.7446 | 0.1093 | 0.1617 |
| Unigram(1.5) | 0.7431 | 0.1102 | 0.1656 |

Table 24: Detectability for Finance QA with SpiritLM

| Method | TPR@FPR | | Median $p$-value |
|---|---|---|---|
| | 1% | 0.1% | |
| KGW($\delta$=1.0) | 0.71 | 0.49 | 0.001 |
| KGW($\delta$=1.5) | 0.94 | 0.79 | 1.1e-05 |
| KGW($\delta$=2.0) | 0.97 | 0.92 | 3.2e-07 |
| Unigram($\delta$=1.0) | 0.19 | 0.06 | 0.151 |
| Unigram($\delta$=1.5) | 0.32 | 0.15 | 0.039 |
| Unigram($\delta$=2.0) | 0.66 | 0.39 | 0.002 |
| DiPmark($\alpha$=0.3) | 0.35 | 0.18 | 0.027 |
| DiPmark($\alpha$=0.4) | 0.48 | 0.23 | 0.012 |
| $\gamma$-reweight | 0.56 | 0.34 | 0.005 |
| ITS | 0.85 | 0.76 | 2.0e-04 |
| ALIGNED-IS | 0.94 | 0.79 | 2.4e-05 |

Table 25: Robustness comparison of watermarking methods for Finance QA with SpiritLM under signal processing attacks. We report TPR at 1% FPR.

| | Watermark | No attack | Echo (0.05sec) | Gauss. noise (30dB) | Lowpass (40%) | Smooth (6 samp.) | Speed (0.9) | Speed (1.1) |
|---|---|---|---|---|---|---|---|---|
| Dist. | KGW($\delta$=1.0) | 0.71 | 0.70 | 0.79 | 0.74 | 0.74 | 0.12 | 0.14 |
| | KGW($\delta$=1.5) | 0.94 | 0.94 | 0.95 | 0.94 | 0.94 | 0.27 | 0.25 |
| | KGW($\delta$=2.0) | 0.97 | 0.98 | 0.98 | 0.97 | 0.98 | 0.37 | 0.36 |
| | Uni($\delta$=1.0) | 0.19 | 0.16 | 0.04 | 0.20 | 0.13 | 0.03 | 0.01 |
| | Uni($\delta$=1.5) | 0.32 | 0.32 | 0.14 | 0.30 | 0.22 | 0.06 | 0.05 |
| | Uni($\delta$=2.0) | 0.66 | 0.62 | 0.39 | 0.66 | 0.57 | 0.17 | 0.12 |
| Dist. free | $\gamma$-reweight | 0.56 | 0.52 | 0.38 | 0.53 | 0.58 | 0.07 | 0.07 |
| | DiP($\alpha$=0.3) | 0.35 | 0.38 | 0.26 | 0.40 | 0.38 | 0.06 | 0.05 |
| | DiP($\alpha$=0.4) | 0.48 | 0.45 | 0.34 | 0.47 | 0.43 | 0.10 | 0.08 |
| | ALIGNED-IS | 0.94 | 0.93 | 0.90 | 0.96 | 0.93 | 0.39 | 0.30 |

ALIGNED-IS outperformed the biased watermarking algorithm KGW and Unigram in most cases, and achieved comparable performance with the strongly biased KGW($\delta$=2.0).

### F.2 Robustness

We supplement the robustness evaluation with Tables 25 through 28 for SpiritLM, and Tables 29 through 34 for SpeechGPT. ALIGNED-IS consistently exhibits the strongest robustness, outperforming all distortion-free watermarking baselines in reliably detecting watermarked audio under adversarial conditions.

## G Broader Impact

We introduce ALIGNED-IS, a distortion-free watermarking framework for autoregressive audio generation models, addressing the retokenization mismatch that limits traditional methods. Beyond the specific technical advancement, its broader impact lies in enhancing the security and trustworthiness of AI-generated audio, which is increasingly critical as synthetic media proliferates. Watermarking technologies such as this not only help identify AI-generated content but also have wider applications, including copyright protection (Liu et al., 2025b; Zhang et al., 2025; Chen et al., 2024a), content authenticity verification, and digital rights management. However, the paper also highlights that robustness remains a key challenge (An et al., 2025; Liu et al., 2025a; Chen et al., 2025b). Watermarks can be degraded or removed through transformations or noise, emphasizing the ongoing need for more resilient and standardized approaches across modalities

Table 26: Robustness comparison of watermarking methods for Finance QA with SpiritLM under codec-based, quantizing, and denoising attacks. We report TPR at 1% FPR.

| | Watermark | EnCodec (24kHz) | MP3 (32kbps) | MP3 (40kbps) | Opus (16kbps) | Opus (31kbps) | Quant. (64-bit) | Denoise |
|---|---|---|---|---|---|---|---|---|
| Dist. | KGW($\delta$=1.0) | 0.69 | 0.54 | 0.51 | 0.69 | 0.73 | 0.64 | 0.67 |
| | KGW($\delta$=1.5) | 0.93 | 0.79 | 0.81 | 0.88 | 0.94 | 0.90 | 0.91 |
| | KGW($\delta$=2.0) | 0.99 | 0.91 | 0.92 | 0.94 | 0.98 | 0.95 | 0.97 |
| | Uni($\delta$=1.0) | 0.10 | 0.14 | 0.14 | 0.12 | 0.17 | 0.01 | 0.08 |
| | Uni($\delta$=1.5) | 0.20 | 0.30 | 0.30 | 0.24 | 0.33 | 0.05 | 0.19 |
| | Uni($\delta$=2.0) | 0.52 | 0.64 | 0.61 | 0.54 | 0.63 | 0.18 | 0.48 |
| Dist. free | $\gamma$-reweight | 0.48 | 0.36 | 0.38 | 0.47 | 0.55 | 0.56 | 0.53 |
| | DiP($\alpha$=0.3) | 0.32 | 0.27 | 0.27 | 0.34 | 0.37 | 0.41 | 0.36 |
| | DiP($\alpha$=0.4) | 0.43 | 0.35 | 0.38 | 0.50 | 0.42 | 0.54 | 0.49 |
| | ALIGNED-IS | 0.91 | 0.81 | 0.80 | 0.92 | 0.93 | 0.85 | 0.89 |

Table 27: Robustness comparison of watermarking methods for LibriSpeech with SpiritLM under signal processing attacks. We report TPR at 1% FPR.

| | Watermark | No attack | Echo (0.05sec) | Gauss. noise (30dB) | Lowpass (40%) | Smooth (6 samp.) | Speed (0.9) | Speed (1.1) |
|---|---|---|---|---|---|---|---|---|
| Dist. | KGW($\delta$=1.0) | 0.69 | 0.52 | 0.64 | 0.55 | 0.58 | 0.05 | 0.09 |
| | KGW($\delta$=1.5) | 0.97 | 0.86 | 0.92 | 0.90 | 0.91 | 0.17 | 0.18 |
| | KGW($\delta$=2.0) | 0.99 | 0.97 | 0.97 | 0.97 | 0.97 | 0.24 | 0.26 |
| | Uni($\delta$=1.0) | 0.06 | 0.05 | 0.01 | 0.05 | 0.04 | 0.01 | 0.00 |
| | Uni($\delta$=1.5) | 0.25 | 0.18 | 0.07 | 0.17 | 0.14 | 0.02 | 0.01 |
| | Uni($\delta$=2.0) | 0.48 | 0.39 | 0.18 | 0.38 | 0.32 | 0.04 | 0.03 |
| Dist. free | $\gamma$-reweight | 0.71 | 0.66 | 0.57 | 0.66 | 0.71 | 0.21 | 0.13 |
| | DiP($\alpha$=0.3) | 0.60 | 0.57 | 0.40 | 0.56 | 0.61 | 0.17 | 0.13 |
| | DiP($\alpha$=0.4) | 0.69 | 0.66 | 0.54 | 0.69 | 0.71 | 0.18 | 0.15 |
| | ALIGNED-IS | 0.95 | 0.92 | 0.90 | 0.93 | 0.92 | 0.28 | 0.21 |

Table 28: Robustness comparison of watermarking methods for LibriSpeech with SpiritLM under codec-based, quantizing, and denoising attacks. We report TPR at 1% FPR.

| | Watermark | EnCodec (24kHz) | MP3 (32kbps) | MP3 (40kbps) | Opus (16kbps) | Opus (31kbps) | Quant. (64-bit) | Denoise |
|---|---|---|---|---|---|---|---|---|
| Dist. | KGW($\delta$=1.0) | 0.45 | 0.29 | 0.31 | 0.51 | 0.53 | 0.47 | 0.53 |
| | KGW($\delta$=1.5) | 0.84 | 0.64 | 0.61 | 0.82 | 0.90 | 0.78 | 0.85 |
| | KGW($\delta$=2.0) | 0.95 | 0.84 | 0.83 | 0.95 | 0.97 | 0.93 | 0.96 |
| | Uni($\delta$=1.0) | 0.03 | 0.05 | 0.04 | 0.03 | 0.04 | 0.00 | 0.02 |
| | Uni($\delta$=1.5) | 0.11 | 0.15 | 0.14 | 0.16 | 0.17 | 0.02 | 0.11 |
| | Uni($\delta$=2.0) | 0.25 | 0.27 | 0.29 | 0.27 | 0.39 | 0.06 | 0.24 |
| Dist. free | $\gamma$-reweight | 0.61 | 0.53 | 0.57 | 0.65 | 0.69 | 0.65 | 0.67 |
| | DiP($\alpha$=0.3) | 0.50 | 0.41 | 0.45 | 0.53 | 0.55 | 0.54 | 0.52 |
| | DiP($\alpha$=0.4) | 0.59 | 0.52 | 0.57 | 0.66 | 0.70 | 0.64 | 0.66 |
| | ALIGNED-IS | 0.90 | 0.75 | 0.80 | 0.89 | 0.91 | 0.85 | 0.87 |

Table 29: Robustness comparison of watermarking methods for Dolly CW with SpeechGPT under signal processing attacks. We report TPR at 1% FPR.

| | Watermark | No attack | Echo (0.05sec) | Gauss. noise (30dB) | Lowpass (40%) | Smooth (6 samp.) | Speed (0.9) | Speed (1.1) |
|---|---|---|---|---|---|---|---|---|
| Dist. | KGW($\delta$=1.0) | 0.32 | 0.28 | 0.28 | 0.31 | 0.29 | 0.03 | 0.03 |
| | KGW($\delta$=1.5) | 0.56 | 0.49 | 0.48 | 0.57 | 0.58 | 0.06 | 0.04 |
| | KGW($\delta$=2.0) | 0.73 | 0.66 | 0.67 | 0.73 | 0.73 | 0.10 | 0.05 |
| | Uni($\delta$=1.0) | 0.17 | 0.19 | 0.16 | 0.17 | 0.18 | 0.12 | 0.09 |
| | Uni($\delta$=1.5) | 0.40 | 0.44 | 0.32 | 0.40 | 0.43 | 0.29 | 0.17 |
| | Uni($\delta$=2.0) | 0.57 | 0.57 | 0.55 | 0.57 | 0.58 | 0.39 | 0.24 |
| Dist. free | $\gamma$-reweight | 0.37 | 0.21 | 0.32 | 0.34 | 0.31 | 0.02 | 0.02 |
| | DiP($\alpha$=0.3) | 0.23 | 0.15 | 0.19 | 0.23 | 0.21 | 0.02 | 0.02 |
| | DiP($\alpha$=0.4) | 0.29 | 0.19 | 0.30 | 0.28 | 0.29 | 0.01 | 0.00 |
| | ALIGNED-IS | 0.82 | 0.78 | 0.81 | 0.81 | 0.80 | 0.23 | 0.15 |

Table 30: Robustness comparison of watermarking methods for Dolly CW with SpeechGPT under codec-based and quantizing attacks. We report TPR at 1% FPR.

| | Watermark | EnCodec (24kHz) | MP3 (32kbps) | MP3 (40kbps) | Opus (16kbps) | Opus (31kbps) | Quant. (64-bit) |
|---|---|---|---|---|---|---|---|
| Dist. | KGW($\delta$=1.0) | 0.26 | 0.16 | 0.16 | 0.29 | 0.30 | 0.14 |
| | KGW($\delta$=1.5) | 0.52 | 0.26 | 0.26 | 0.56 | 0.56 | 0.36 |
| | KGW($\delta$=2.0) | 0.67 | 0.45 | 0.43 | 0.69 | 0.71 | 0.53 |
| | Uni($\delta$=1.0) | 0.17 | 0.15 | 0.15 | 0.19 | 0.18 | 0.06 |
| | Uni($\delta$=1.5) | 0.35 | 0.28 | 0.29 | 0.41 | 0.40 | 0.12 |
| | Uni($\delta$=2.0) | 0.55 | 0.49 | 0.46 | 0.61 | 0.60 | 0.23 |
| Dist. free | $\gamma$-reweight | 0.32 | 0.16 | 0.18 | 0.31 | 0.35 | 0.23 |
| | DiP($\alpha$=0.3) | 0.22 | 0.10 | 0.10 | 0.19 | 0.24 | 0.12 |
| | DiP($\alpha$=0.4) | 0.27 | 0.12 | 0.14 | 0.24 | 0.30 | 0.20 |
| | ALIGNED-IS | 0.78 | 0.64 | 0.60 | 0.78 | 0.82 | 0.62 |

Table 31: Robustness comparison of watermarking methods for Finance QA with SpeechGPT under signal processing attacks. We report TPR at 1% FPR.

| | Watermark | No attack | Echo (0.05sec) | Gauss. noise (30dB) | Lowpass (40%) | Smooth (6 samp.) | Speed (0.9) | Speed (1.1) |
|---|---|---|---|---|---|---|---|---|
| Dist. | KGW($\delta$=1.0) | 0.36 | 0.30 | 0.34 | 0.37 | 0.35 | 0.05 | 0.03 |
| | KGW($\delta$=1.5) | 0.68 | 0.65 | 0.69 | 0.70 | 0.71 | 0.07 | 0.07 |
| | KGW($\delta$=2.0) | 0.80 | 0.78 | 0.78 | 0.81 | 0.83 | 0.10 | 0.07 |
| | Uni($\delta$=1.0) | 0.21 | 0.27 | 0.14 | 0.19 | 0.21 | 0.20 | 0.15 |
| | Uni($\delta$=1.5) | 0.64 | 0.70 | 0.56 | 0.60 | 0.64 | 0.40 | 0.31 |
| | Uni($\delta$=2.0) | 0.72 | 0.72 | 0.71 | 0.74 | 0.76 | 0.52 | 0.39 |
| Dist. free | $\gamma$-reweight | 0.56 | 0.37 | 0.51 | 0.52 | 0.55 | 0.03 | 0.04 |
| | DiP($\alpha$=0.3) | 0.33 | 0.26 | 0.39 | 0.39 | 0.33 | 0.01 | 0.00 |
| | DiP($\alpha$=0.4) | 0.54 | 0.36 | 0.49 | 0.51 | 0.48 | 0.02 | 0.03 |
| | ALIGNED-IS | 0.94 | 0.94 | 0.94 | 0.94 | 0.93 | 0.38 | 0.24 |

Table 32: Robustness comparison of watermarking methods for Finance QA with SpeechGPT under codec-based and quantizing attacks. We report TPR at 1% FPR.

| | Watermark | EnCodec (24kHz) | MP3 (32kbps) | MP3 (40kbps) | Opus (16kbps) | Opus (31kbps) | Quant. (64-bit) |
|---|---|---|---|---|---|---|---|
| Dist. | KGW($\delta$=1.0) | 0.36 | 0.17 | 0.18 | 0.35 | 0.39 | 0.16 |
| | KGW($\delta$=1.5) | 0.69 | 0.38 | 0.40 | 0.71 | 0.69 | 0.50 |
| | KGW($\delta$=2.0) | 0.75 | 0.58 | 0.61 | 0.80 | 0.82 | 0.66 |
| | Uni($\delta$=1.0) | 0.21 | 0.16 | 0.15 | 0.23 | 0.25 | 0.05 |
| | Uni($\delta$=1.5) | 0.61 | 0.50 | 0.51 | 0.71 | 0.66 | 0.22 |
| | Uni($\delta$=2.0) | 0.71 | 0.63 | 0.65 | 0.77 | 0.75 | 0.44 |
| Dist. free | $\gamma$-reweight | 0.52 | 0.32 | 0.30 | 0.47 | 0.56 | 0.37 |
| | DiP($\alpha$=0.3) | 0.38 | 0.19 | 0.19 | 0.31 | 0.36 | 0.23 |
| | DiP($\alpha$=0.4) | 0.45 | 0.28 | 0.26 | 0.47 | 0.49 | 0.31 |
| | ALIGNED-IS | 0.91 | 0.89 | 0.88 | 0.95 | 0.94 | 0.87 |

Table 33: Robustness comparison of watermarking methods for Longform QA with SpeechGPT under signal processing attacks. We report TPR at 1% FPR.

| | Watermark | No attack | Echo (0.05sec) | Gauss. noise (30dB) | Lowpass (40%) | Smooth (6 samp.) | Speed (0.9) | Speed (1.1) |
|---|---|---|---|---|---|---|---|---|
| Dist. | KGW($\delta$=1.0) | 0.40 | 0.38 | 0.38 | 0.44 | 0.38 | 0.06 | 0.01 |
| | KGW($\delta$=1.5) | 0.72 | 0.57 | 0.59 | 0.75 | 0.63 | 0.08 | 0.04 |
| | KGW($\delta$=2.0) | 0.84 | 0.83 | 0.81 | 0.83 | 0.83 | 0.10 | 0.08 |
| | Uni($\delta$=1.0) | 0.25 | 0.30 | 0.22 | 0.23 | 0.29 | 0.15 | 0.10 |
| | Uni($\delta$=1.5) | 0.55 | 0.61 | 0.48 | 0.54 | 0.60 | 0.38 | 0.27 |
| | Uni($\delta$=2.0) | 0.71 | 0.75 | 0.67 | 0.72 | 0.71 | 0.53 | 0.32 |
| Dist. free | $\gamma$-reweight | 0.46 | 0.26 | 0.39 | 0.48 | 0.41 | 0.04 | 0.02 |
| | DiP($\alpha$=0.3) | 0.37 | 0.25 | 0.33 | 0.32 | 0.36 | 0.04 | 0.03 |
| | DiP($\alpha$=0.4) | 0.46 | 0.33 | 0.40 | 0.46 | 0.37 | 0.01 | 0.01 |
| | ALIGNED-IS | 0.95 | 0.93 | 0.93 | 0.95 | 0.92 | 0.38 | 0.22 |

Table 34: Robustness comparison of watermarking methods for Longform QA with SpeechGPT under codec-based and quantizing attacks. We report TPR at 1% FPR.

| | Watermark | EnCodec (24kHz) | MP3 (32kbps) | MP3 (40kbps) | Opus (16kbps) | Opus (31kbps) | Quant. (64-bit) |
|---|---|---|---|---|---|---|---|
| Dist. | KGW($\delta$=1.0) | 0.40 | 0.21 | 0.17 | 0.37 | 0.42 | 0.19 |
| | KGW($\delta$=1.5) | 0.63 | 0.38 | 0.42 | 0.63 | 0.70 | 0.41 |
| | KGW($\delta$=2.0) | 0.80 | 0.59 | 0.61 | 0.84 | 0.84 | 0.64 |
| | Uni($\delta$=0.5) | 0.07 | 0.05 | 0.03 | 0.08 | 0.08 | 0.04 |
| | Uni($\delta$=1.0) | 0.21 | 0.20 | 0.21 | 0.29 | 0.24 | 0.05 |
| | Uni($\delta$=1.5) | 0.54 | 0.49 | 0.46 | 0.60 | 0.58 | 0.14 |
| | Uni($\delta$=2.0) | 0.69 | 0.61 | 0.64 | 0.77 | 0.73 | 0.35 |
| Dist. free | $\gamma$-reweight | 0.40 | 0.26 | 0.25 | 0.38 | 0.48 | 0.30 |
| | DiP($\alpha$=0.3) | 0.34 | 0.20 | 0.16 | 0.26 | 0.37 | 0.17 |
| | DiP($\alpha$=0.4) | 0.43 | 0.19 | 0.16 | 0.37 | 0.37 | 0.27 |
| | ALIGNED-IS | 0.91 | 0.82 | 0.84 | 0.95 | 0.96 | 0.85 |

