# OpenReview forum: "Robust Distortion-Free Watermark for Autoregressive Audio Generation Models"
_NeurIPS.cc/2025/Conference — NeurIPS 2025 poster_

### Official Review · Reviewer_pf3V · 2025-06-18

**Clarity:** 2
**Significance:** 2
**Originality:** 3
**Rating:** 4
**Confidence:** 4

**Summary:**

This paper find retokenization mismatch phenomenon when applying the previous LM watermarking algorithms to autoregressive audio generation models. To address this issue, they propose ALIGNED-IS with audio token clustering and aligned inverse sampling. This method allows add distortion-free watermark to generated audio with outstanding detectability and robustness to audio coding. However, there are some concerns about the adequacy of the experiments and the statement of limitations. Please refer to the Weakness Questions, and Limitations.

**Questions:**

1. It may not be appropriate to use FAD alone to demonstrate audio quality, which, as it is defined, measures only divergence between two distributions.  Meanwhile, authors stated that “audios processed by WavMark have audible high-frequency artifacts” in Appendix C.2. However, they obtain higher NISQA and DNSMOSPro, as shown in Tables 14, 15, 16, and 17. Could authors clarify the potential reasons for this discrepancy and provide more experimental results to demonstrate that “post hoc audio watermarking degrade output quality”. Moreover, I’m wondering about NISQA and DNSMOSPro scores for other baselines in Figure 2.
2. If I understand correctly, ALIGNED-IS can only convey information like "Yes" or "No". It raises the limitation that its information capacity is inferior to post-processing methods.
3. According to the results in Appendix C, ALIGNED-IS is not robust to speed changes. It could be helpful to discuss this for a full presentation of this work.
4. In section 3.1, the authors stated “Each code θi is typically derived from a secret key key ∈ K and the n-gram preceding context, denoted xt−n:t−1” Could you clarify this with instance or references?

**Ethical Concerns:**

["NO or VERY MINOR ethics concerns only"]

**Final Justification:**

The rebuttal provided by the authors has addressed my concerns well, and I have decided to maintain my positive evaluation.

**Limitations:**

I suggest incorporating a discussion of the limitations of application, information capacity, and robustness into Appendix A.

**Paper Formatting Concerns:**

No formatting issues.

**Quality:**

3

**Strengths And Weaknesses:**

Strengths:
1. The “audio token clustering” substantially mitigates the retokenization mismatch problem.
2. The aligned inverse sampling can generate distortion-free, detectable, and robust watermarks for autoregressive audio generation models.
Weaknesses:
1. If I understand correctly, ALIGNED-IS is only applicable to audio generation models based on LM and discrete audio tokens. They are challenging to apply to other audio generation models, such as those based on LM and diffusion, as well as those based on diffusion only, which are also popular audio generation methods.
2. The evaluations of audio quality seem insufficient. The detailed comment is given in the Questions.
3. The discussion of limitations is not comprehensive enough.

---

> ### Author Rebuttal · Authors · 2025-07-29
>
> Thank you for your positive and thoughtful feedback. We're glad you found the audio token clustering effective in addressing retokenization mismatch, and that the aligned inverse sampling approach successfully balances distortion-free generation, detectability, and robustness.
>
> > W1. If I understand correctly, ALIGNED-IS is only applicable to audio generation models based on LM and discrete audio tokens. They are challenging to apply to other audio generation models, such as those based on LM and diffusion, as well as those based on diffusion only, which are also popular audio generation methods.
>
> **A1.** Yes. ALIGNED‑IS is intentionally scoped to LM‑based, discrete‑token audio generators, a family that is widely deployed for high‑quality, low‑latency speech synthesis. The only structural requirement our method uses is a sequential stream of discrete symbols. Under that condition, the same cluster‑level watermarking mechanism carries over to any model that generates discrete tokens incrementally, including LM–diffusion hybrids that quantize audio with VQ‑codebooks (e.g., EnCodec/SoundStream‑style front‑ends) and transducer/RNN‑T style decoders. For pure diffusion pipelines that never expose discrete tokens, two straightforward adaptations exist: a) Discretize an intermediate latent via a learned codebook and apply our cluster‑level watermark there; or b) Treat each diffusion step as a pseudo time index and watermark step‑wise cluster assignments in the latent space.
>
> > W2. The discussion of limitations is not comprehensive enough.
>
> **A2.** We will refine the limitation discussion based on the reviewers’ comments and our rebuttal.
>
> > Q1. It may not be appropriate to use FAD alone to demonstrate audio quality, which, as it is defined, measures only divergence between two distributions. Meanwhile, authors stated that “audios processed by WavMark have audible high-frequency artifacts” in Appendix C.2. However, they obtain higher NISQA and DNSMOSPro, as shown in Tables 14, 15, 16, and 17. Could authors clarify the potential reasons for this discrepancy and provide more experimental results to demonstrate that “post hoc audio watermarking degrades output quality”. Moreover, I’m wondering about NISQA and DNSMOSPro scores for other baselines in Figure 2.
>
> **A.** Thank you for raising this point. While WavMark achieves higher scores on certain automated MOS metrics like NISQA and DNSMOSPro, this is likely because it is explicitly optimized for those metrics, which do not always align with perceptual audio quality, especially when artifacts such as high-frequency noise are introduced, as noted in Appendix C.2. Since automated "MOS" scores can be difficult to interpret meaningfully, we follow [1] and Reviewer Hewn's suggestion to include additional quality metrics such as speaker similarity, ASR-CER (character error rate), and ASR-WER (word error rate) on SpiritLM with Librispeech dataset. Across these dimensions, ALIGNED-IS consistently outperforms both biased watermarking methods (e.g., KGW, Unigram) and post-hoc watermarking approaches like WavMark, providing more robust and interpretable evidence of its superior audio fidelity in practice.
>
> Speaker Similarity: WavLM-Base [6].
>
> ASR-CER/WER: Hubert-large, fine-tuned [7].
> ,
> | Method      | Speaker Similarity ↑ | ASR-CER ↓ | ASR-WER ↓ |
> |-------------|-----------------------|-----------|-----------|
> | Baseline    | 0.7550                | 0.1039    | 0.1475    |
> | Audioseal   | 0.7479                | 0.1086    | 0.1648    |
> | Wavmark     | 0.7311                | 0.1143    | 0.1819    |
> | Aligned-IS  | 0.7548                | 0.1038    | 0.1472    |
> | DiPmark(0.4)  | 0.7543                | 0.1045    | 0.1468   |
> | \gamma-reweight| 0.7546                | 0.1043    | 0.1477    |
> | KGW(1.5) | 0.7446                | 0.1093    |  0.1617    |
> | Unigram(1.5)  | 0.7431                | 0.1102    | 0.1656    |
>
> > Q2. If I understand correctly, ALIGNED-IS can only convey information like "Yes" or "No". It raises the limitation that its information capacity is inferior to post-processing methods.
>
> **A.** Yes, the current algorithm focuses on zero-bit watermarks. To improve the information capacity (e.g. n-bit watermarking), one can a) get the bit expression e.g. (1011…1), b) create n+1 ALIGNED-IS which will be sequentially executed on the output token logits, c) if the i-th bit is 1, we apply the i+1-th ALIGNED-IS, if not, we skip this step. During detection, we can recover the i-th bit by the detecting results of the i+1-th ALIGNED-IS (1 for watermarked and 0 otherwise).
>
> > Q3. According to the results in Appendix C, ALIGNED-IS is not robust to speed changes. It could be helpful to discuss this for a full presentation of this work.
>
> **A.** We agree that ALIGNED-IS, like other methods relying on fixed-frame-rate audio tokenizers, is currently not robust to speed changes. Speed perturbations introduce temporal drift between the original and modified token sequences, which can disrupt cluster assignments and degrade detection. This is a known limitation of token-based watermarking schemes under desynchronization attacks. However, we believe this issue is addressable: for example, by applying a grid search to estimate and correct for the speed change, we can potentially realign the modified audio back to its original tokenization frame rate. We will include a discussion of this limitation and possible mitigation strategies in the revised manuscript to provide a more complete picture of the method's strengths and limitations.
>
> > Q4. In section 3.1, the authors stated “Each code θi is typically derived from a secret key key ∈ K and the n-gram preceding context, denoted xt−n:t−1” Could you clarify this with instance or references?
>
> **A.** The sentence refers to the standard key derivation process used in many prior watermarking schemes, e.g., [2–5], where the watermark code (or seed) is computed from a fixed secret key and the preceding n-gram context. For example, if we set $n=2$ (2-gram) and the preceding tokens are $\texttt{[Hu221]}, \texttt{[Hu105]}, \texttt{[Hu142]}$, and we aim to generate the next token after $\texttt{[Hu142]}$, the corresponding watermark code would be $(k, \texttt{[Hu105]}~\texttt{[Hu142]})$, where $k$ is a fixed secret key such as $k=(111000\ldots1)$. This design allows the watermarking scheme to inject token-level perturbations in a context-dependent but reproducible way, as adopted in [2–5]. We will clarify this with an example and references in the revised manuscript.
>
> > L: I suggest incorporating a discussion of the limitations of application, information capacity, and robustness into Appendix A.
>
> **A.** We will add such discussion following the reviewer’s suggestion and the rebuttal.
>
> [1]  O’Reilly et al. Deep Audio Watermarks are Shallow: Limitations of Post-Hoc Watermarking Techniques for Speech. ICLR GenAI workshop.
>
> [2] A watermark for large language models, ICML 2023
>
> [3] Unbiased watermark for large language model, ICLR 2024
>
> [4] DiPmark: A Stealthy, Efficient and Resilient Watermark for Large Language Models, ICML 2024.
>
> [5] Dathathri et al. Scalable watermarking for identifying large language model 395 outputs. Nature 2024.
>
> [6] Chen et al. Wavlm: Large-scale self-supervised pre-training for full stack speech processing, 2022.
>
> [7] Hsu et al. Hubert: Self-supervised speech representation learning by masked prediction of hidden units, 2021.

---

> > ### Comment · Reviewer_pf3V · 2025-08-04
> >
> > Thank you for the clarification. I will keep my rating.

---

> > > ### Author Response · Authors · 2025-08-04
> > > **Thank you!**
> > >
> > > We’re glad our response resolved your concerns and appreciate your positive rating.

---

### Official Review · Reviewer_Hewn · 2025-06-27

**Clarity:** 3
**Significance:** 3
**Originality:** 3
**Rating:** 5
**Confidence:** 4

**Summary:**

The authors propose a language model-style watermarking scheme for autoregressive generative models operating on audio tokens. Existing approaches struggle due to the fact that audio tokenizers lack cycle consistency / idempotence, meaning that decoding tokens to audio and re-encoding results in substantial changes to the tokens, thereby thwarting existing statistical watermark detection schemes. The proposed method uses clustering based on pre-quantization encoded features to circumvent this issue. The authors demonstrate that via an Aligned ITS scheme, the proposed method can generate robustly detectable watermarked audio by guiding sampling at the cluster level rather than the token level.

**Questions:**

The paper checklist seems to indicate that code is provided in a supplement, but I can not find any supplementary material beyond the paper's appendix.

The authors claim 0% detection rates for AudioSeal under processing with the original Encodec, which is surprising given the Encodec experiments in the original AudioSeal paper and the results from O'Reilly et al. (https://arxiv.org/abs/2504.10782). The 12kHz bandwidth seems unlikely to have an effect here as AudioSeal has an 8kHz bandwidth (16kHz sample rate), so I'm wondering if there might be a bug? Or perhaps the 12kHz Encodec is just much more effective at removing watermarks than the full-bandwidth version.

**Ethical Concerns:**

["NO or VERY MINOR ethics concerns only"]

**Final Justification:**

The authors addressed the majority of my concerns in the rebuttal, including improved audio quality metrics, robustness evaluation, and desynchronization evaluation. As a result, I have raised my score.

**Limitations:**

Yes.

**Paper Formatting Concerns:**

None.

**Quality:**

3

**Strengths And Weaknesses:**

__Strengths__

* Adapting "semantic" / "generative" / "intrinsic" watermarks from the text modality to other modalities such as audio is an important research direction -- particularly in audio, where imperceptible post-hoc watermarks currently dominate but have been shown to be fragile against re-generation attacks.

* The proposed method does not require training a robust tokenizer to circumvent the "retokenization mismatch" issue, and is therefore widely applicable to existing autoregressive audio generative models

* The robustness evaluations indicate the proposed method may improve on the robustness of post-hoc audio watermarks under certain classes of transformation

* The proposed Aligned-ITS scheme is interesting, and detection only relies on the assumption that the conditional probabilities of token clusters are approximately equal ($\approx \frac{1}{h}$)



__Weaknesses__

* The authors do not appear to mention the length of generated audio required to detect the watermark. The authors need to clarify the average length of generated audio used in their detection experiments, and ideally conduct an additional experiment demonstrating detection performance as a function of generation length. Readers will want to know: "Can I detect this watermark in a 5-second random excerpt of generated audio? In a 2-second excerpt?"

* While the proposed watermark is "distortion free" and should in theory produce high quality audio, the audio quality evaluations are unconvincing:
  * The use of FAD is suspect. AudioSeal performs poorly on FAD compared to the proposed method despite being essentially human-imperceptible, raising questions about the alignment of this metric with human perception. The authors do not state what underlying embedding model is used to compute FAD. If these embeddings are computed on e.g. SSL features at a fixed small hop length, then note that a "watermark" that chops and randomly re-shuffles audio could achieve near-0 FAD because the underlying frame-level embedding sets would be the same, despite the watermark destroying audio quality.
  * The authors do not use standard metrics for evaluating the output of autoregressive speech language models such as ASR word/character error rates (for intelligibility) and speaker similarity (if a speaker audio prompt is provided). These are much more interpretable than automated "MOS" evaluations, and basically standard -- for instance, the paper for the SpiritLM model used by the authors evaluates ASR CER, the VALL-E paper evaluates ASR WER and speaker similarity, etc.
  * The authors do not conduct a human listening evaluation or provide audio examples of their watermark. This makes it difficult to trust that the proposed method does not noticeably deteriorate generated audio quality. The authors should certainly provide listening examples

* The robustness of the proposed method against desynchronization is not clear. It performs most poorly against speed change, which indicates a potential issue in which a fixed-frame-rate audio tokenizer results in "drift" between token streams from the original generated audio and generated audio that has been shifted in time, time-stretched, or had its speed changed. The authors could address this with some very simple experiments:
  * Akin to Figure 7 in the WavMark paper, evaluate detection performance as a function of time shift (but include both positive and negative shifts). This should address the concern over whether small shift of e.g. 25% or 50% of the tokenizer frame length results in tokens being mapped to a different cluster (thus destroying the watermark)
  * For different speed change amounts, plot token-level detection scores as a function of token sequence position. As we move farther along in the watermarked audio, do our detection scores fall off as the speed change causes the detector to desynchronize more and more?

* The authors only evaluate robustness to one codec-based regeneration attack (the original Encodec). Works such as O'Reilly et al. (https://arxiv.org/abs/2504.10782) have shown this to be much weaker at removing watermarks than recent low-bitrate codecs. It would nice to see the authors evaluate a couple of more recent low-bitrate codec attacks, as this is where even stronger post-hoc watermarks not considered by the authors (e.g. Timbre-Watermark) fail.

* The general robustness of many watermarking methods evaluated by the authors (e.g. KGW, ITS) even without the proposed clustering mechanism seems to suggest that a significant portion of the observed robustness comes from the robustness of the tokenizers used rather than the clustering mechanism.

* Limited scope: despite the potential generality of the proposed method, the authors only evaluate their method within two multimodal speech LLMs. This approach should in theory be applicable to TTS models (VALL-E, Parler-TTS) music generation models (MusicGen), etc. While the authors do not consider multi-level RVQ audio tokenizers, which still dominate many audio generation tasks, the proposed method should in theory still be applicable (e.g. by watermarking only the first RVQ level). Overall, I think this weakness is minor compared to the others mentioned above, and probably not feasible to address in the rebuttal period.

* Related work:
  * The "retokenization mismatch" referred to by the authors, including its relevance to language model-style watermarking techniques, has been covered by Défossez et al. (https://arxiv.org/abs/2410.00037, arXiv) and by O'Reilly et al. (https://arxiv.org/abs/2410.11025, ICASSP 2025).

  * The authors do not address previous attempts at discrete token-based audio watermarking (e.g. "DiscreteWM", AAAI 2025 https://arxiv.org/abs/2412.13917) and "generative" or "intrinsic" approaches that attempt to ground watermarks within generative models (e.g. "GROOT", ACM MM 2024, https://arxiv.org/abs/2407.10471). While these existing approaches have potential robustness limitations (DiscreteWM) or produce "shallow" signatures akin to post-hoc watermarks (GROOT), they definitely merit a mention. Among attempted "generative" or "intrinsic" audio watermarks, the following publications come to mind:

    * https://arxiv.org/abs/2309.15224 (ICASSP 2024)
    * https://arxiv.org/abs/2409.13382 (ICASSP 2025)
    * https://arxiv.org/abs/2407.10471 (ASM MM 2024)
    * https://arxiv.org/abs/2406.04840 (Interspeech 2024)
    * https://arxiv.org/abs/2202.08900 (ICASSP 2022)


__Summary and Recommendations__

As a result of the above concerns, I do not currently recommend accepting the paper. __However, I am willing to raise my score provided the authors address the most pressing points mentioned above__. In order of significance, these are:
1. The lack of audio examples and flawed audio quality evaluation (audio examples most important, followed by standard quality metrics)
2. The lack of information on the required detection length
3. Potential synchronization issue
4. Lack of more recent codec attacks in robustness evaluation

Addressing points 1-2 would move my rating to accept. Additionally addressing points 3-4 would provide much stronger evidence that the proposed method is actually superior to post-hoc watermarks against the most challenging attacks, and would further increase my score. Concerns re: related work should be trivial to address, and I'm assuming the authors will add missing citations or otherwise discuss why they feel these prior works are not relevant.

Overall, I think this work has the potential to make a strong case for the feasibility of language model-style "semantic" watermarking of autoregressive audio generative models.

---

> ### Author Rebuttal · Authors · 2025-07-31
>
> Thank you for your very detailed, thoughtful, and constructive feedback. We appreciate your recognition of the broader importance of adapting generative watermarking methods beyond text. Below we address the concerns.
>
> > W1. The length of generated audio required to detect the watermark.
>
> **A1.** The length of audio token sequences used for detection is typically around 500, which corresponds to approximately 10–20 seconds of audio. The duration is affected by the tokenizer and vocoder frame rates. Some variation arises because the duration of each token may differ depending on the model’s duration prediction. In the table below, we present detection results on randomly cropped segments of watermarked audio with varying durations. As shown, detection performance improves with longer audio segments, achieving reliable detectability for audio durations exceeding 5 seconds.
>
> SpeechGPT/Longform QA
>
> | Time (sec) | TPR@FPR=1% |
> |---|--|
> | ~2 |0.50|
> | ~3 |0.67|
> | ~4|0.82|
> | ~5| 0.87|
> | ~6 | 0.92|
> | ~7 | 0.93|
> | ~8| 0.94|
> | ~9| 0.95 |
>
> > W2 a) The use of FAD is suspect.
>
> **A2 a).** FAD it is not explicitly aligned with human perception, the poor results of AudioSeal most likely arise from deviations in barely audible frequencies. In our paper, we used the CNN-based VGGish model to compute FAD [1], which uses non-overlapping frames and would not be robust to the reshuffling attack that the reviewer aptly points out. We thus re-evaluated the audios using the transformer-based CLAP model as the FAD feature extractor and verified our distortion-free claim.
>
> | Method  | FAD (CLAP) |
> |--|--|
> | No watermark   | 0.0036|
> | KGW ($\delta$=2.0) | 0.0052|
> | Uni ($\delta$=2.0) | 0.0041 |
> | $\gamma$-reweight  | 0.0034|
> | AudioSeal  | 0.0084 |
> | WavMark | 0.0693 |
> | AlignedIS | 0.0031 |
>
> > W2 b) Evaluate audio quality under ASR word/character error rates and speaker similarity.
>
> **A2 b).** We evaluate speaker similarity using WavLM-Base [3], and ASR-CER/WER using HuBERT-large fine-tuned model [4], with watermarked audios. From the results, we observe that distortion-free watermarking methods, e.g., Aligned-IS, DiPmark, and γ-reweight, achieve performance comparable to the unwatermarked baseline across all metrics. In contrast, biased and post-hoc watermarking methods noticeably degrade the generation quality.
>
> SpiritLM/Librispeech
> | Method      | Speaker Similarity ↑ | ASR-CER ↓ | ASR-WER ↓ |
> |---|---|--|--|
> | No watermark    | 0.7550 | 0.1039 | 0.1475|
> | Audioseal | 0.7479 | 0.1086| 0.1648 |
> | Wavmark| 0.7311 | 0.1143| 0.1819|
> | Aligned-IS  | 0.7548  | 0.1038| 0.1472|
> | DiPmark(0.4)  | 0.7543 | 0.104 | 0.1468|
> | γ-reweight| 0.7546 | 0.1043| 0.1477 |
> | KGW(1.5) | 0.7446| 0.1093|  0.1617 |
> | Unigram(1.5)  | 0.7431 | 0.1102| 0.1656|
>
>
> > W2 c) The authors do not conduct a human listening evaluation or provide audio examples of their watermark.
>
> **A2 c).** According to the NeurIPS rebuttal guideline, we are not able to add links to the audio examples, but we will definitely provide audio examples in the supplementary if the paper gets accepted. To assess the quality of Aligned-IS, we conducted a human listening study with five English-speaking participants. Each participant was presented with 50 randomly ordered pairs of watermarked and unwatermarked audio samples and asked to select the one with better quality in each pair. A binomial test under the null hypothesis that both versions are equally likely to be preferred ($p = 0.5$) yielded a $p$-value of 0.73, indicating no statistically significant difference. We conclude that Aligned-IS introduces no perceptible degradation in audio quality.
> | Speaker |1|2 |3 |4 |5 | Total |
> |-|-|-|-|--|-|-|
> | Unwatermarked Win|24 |27 |28 |23 | 25|127|
> | Aligned-IS   Win |26|23|22 |27 |25 |12 |
>
>
>
> > W3. The robustness of the proposed method against desynchronization is not clear.
>
> **A3.** We acknowledge that our current watermarking scheme, which relies on a fixed-frame-rate audio tokenizer, is not inherently robust to speed-based desynchronization attacks such as time-stretching or speed perturbation. These attacks introduce temporal "drift" between the original and modified token frames, potentially disrupting cluster assignments and degrading detection performance. However, this limitation can be partially addressed using post-hoc alignment techniques e.g. by applying a grid search to resynchronize the modified audio with the original timing.
>
> > W3 a) Evaluate detection performance as a function of time shift.
>
> **A3 a).** Indeed, the proposed watermark is not inherently robust to time shift, but deception of the watermark would be possible by tokenization at a few various small offsets (smaller than the frame size). We performed an experiment with the Longform QA dataset and the SpiritLM model, whose frame size is 645 samples. We uniformly sampled 8 time shifts, with shifts larger than 322 being equivalent to negative shifts of less than 50% of the frame length. In the following Table, we observe our watermark’s strength decrease and then increase, but still maintaining high detectability at small offsets.
>
> | Offset (samples) | TPR@FPR=1% |
> |---|-|
> | 0   | 0.94|
> | 80 | 0.92|
> | 160| 0.76|
> | 240 | 0.58|
> | 320  | 0.43 |
> | 400 | 0.37|
> | 480| 0.54|
> | 560| 0.88|
> | 640 | 0.94|
>
> > W3 b) Plot token-level detection scores as a function of token sequence position.
>
> **A3 b).** The reviewer’s assumption is very reasonable, but it does not apply to our watermarking scheme because its positive scores are sparsely distributed in the token sequence. The sparsity is controlled by the granularity of clusters used. We plotted the average token-level scores as a function of token position but did not observe a diminishing trend.
>
> > W4. The authors only evaluate robustness to one codec-based regeneration attack (the original Encodec).
>
> **A4.** We evaluate two recent stronger codec-based attacks proposed in O'Reilly et al.: FACodec [5] and SpeechTokenizer [6]. We use the default configurations from their official GitHub repositories, corresponding to 1.6 kbps for FACodec and 4 kbps for SpeechTokenizer. As shown in the results below, Aligned-IS remains effective under both attacks, while Audioseal and WavMark fail to maintain detection performance, verifying O’Reilly et al.’s fragility conclusion.
>
> SpiritLM / Longform QA
> | TPR@FPR=1% |FACodec | SpeechTokenizer|
> |--|--|-|
> | Aligned-IS | 0.95 | 0.92 |
> | Audioseal | 0.00 | 0.00 |
> | Wavmark | 0.00 | 0.00 |
>
> SpeechGPT / Longform QA
> | TPR@FPR=1% |FACodec | SpeechTokenizer|
> |--|--|--|
> | Aligned-IS | 0.93 | 0.91 |
> | Audioseal | 0.00 | 0.00 |
> | Wavmark | 0.00 | 0.00 |
>
> > W5.  A significant portion of the observed robustness comes from the robustness of the tokenizers...
>
> **A5.** While tokenizer stability contributes to robustness, our clustering mechanism targets a distinct challenge: **retokenization mismatch** in discrete audio generation, where decoding variability alters token sequences despite perceptual similarity. Clustering mitigates this issue by grouping similar tokens, making watermark detection more resilient. It complements, rather than replaces, tokenizer robustness by addressing a modality-specific failure mode.
>
> > W6. Limited scope
>
> **A6.** We appreciate the reviewer’s suggestion. However, applying our method to models like VALL-E, Parler-TTS, or MusicGen is nontrivial due to their cross-modal nature, where input and output modalities differ. These models lack a reverse mapping from audio to token space, which is essential for verification. Supporting them would require training an additional audio-to-token module, undermining the training-free nature of our approach.
>
> Despite that, our approach is extensible beyond audio models. For example, autoregressive image generation models such as Emu3 [2] also employ an image-token encoder-decoder structure. We conducted a clustering analysis on Emu3 and observed similar retokenization mismatch patterns, which our clustering method effectively mitigates (see the table below).
>
> | Mismatch Rate Before | Mismatch Rate After | Reduction|
> |---|-|-|
> | 0.5131 | 0.2305 | 55.07% |
>
>
> > W7. Related work:
>
> A7. Thank you for highlighting the missing related work. We will incorporate a detailed discussion of these references in the revised manuscript.
>
> > Q1. Code is missing
>
> **A.** We truly apologize for the missing code. Since we are not allowed to submit links in the rebuttal, we send the anonymous github link to AC according to the NeurIPS 2025 rebuttal guidelines.
>
> > Q2. 0% detection rates for AudioSeal under original Encodec.
>
> **A.** We sincerely thank the reviewer for identifying this issue. Indeed, we examined our implementation and identified that EnCodec wrongly assumed 24 kHz sampling rate, using a hard-coded value. After fixing our code and verifying that the rest of the attacks did not contain similar issues, we re-ran the EnCodec attack under varying bitrate settings. As expected, the performance of WavMark and Audioseal remains poor across all bandwidths. While Audioseal shows an improvement as the bandwidth increases, it still fails to achieve a strong detection rate.
>
> |  Bitrate (kbps) | 3 | 6  | 12 | 24 |
> |--|--|-|--|--|
> | Audioseal detection | 0| 0 | 0| 0.05 |
> | Audioseal avg. probability | 0.3733  | 0.4413  | 0.7252  | 0.9368  |
> | Wavmark detection   | 0 | 0 | 0 | 0 |
> | AlignedIS detection | 0.8740  | 0.8940  | 0.9040  | 0.9160  |
>
> [1] Fréchet Audio Distance: A Reference-free Metric for Evaluating Music Enhancement Algorithms. Interspeech. 2019.
>
> [2] Emu3: Next-Token Prediction is All You Need, arxiv2409.18869.
>
> [3] Wavlm: Large-scale self-supervised pre-training for full stack speech processing, 2022.
>
> [4] Hubert: Self-supervised speech representation learning by masked prediction of hidden units, 2021.
>
> [5] Naturalspeech 3: Zero-shot speech synthesis with factorized codec and diffusion models. arXiv:2403.03100.
>
> [6] Speechtokenizer: Unified speech tokenizer for speech large language models. ICLR 2024.

---

> > ### Comment · Reviewer_Hewn · 2025-08-05
> > **Reply to Authors**
> >
> > I thank the authors for their detailed reply and for running additional experiments.
> >
> > __A1.__ This is an interesting result and should definitely improve the paper. In general, it looks like the proposed method doesn't approach the TPR@1%FPR performance of post-hoc watermarks like AudioSeal until audio lengths near 10 seconds, and presumably underperforms at smaller FPR. This is important information, and shows that there is room for improvement in achieving the kind of low false positive rates necessary for real-world deployment (where false positives are costly).
> >
> > __A2.__ I appreciate the authors running these experiments. In particular, the ASR/ASV metrics improve the argument for audio quality preservation.
> >
> > __A3.__ Again, I appreciate the additional experiments and frank discussion of vulnerability to desynchronization attacks. I think this will strengthen the paper and help alleviate concerns about desynchronization.
> >
> > __A4.__ This is a good result, and a strong argument in favor of the proposed method when considering robustness to sophisticated adversaries.
> >
> > __A5.-A7.__ Also looks reasonable to me.
> >
> > Overall, this is a very thorough rebuttal and addresses many of the concerns mentioned in my review. I have correspondingly increased my score.

---

> > > ### Author Response · Authors · 2025-08-06
> > > **Thank you!**
> > >
> > > We sincerely thank you for your thoughtful feedback, which has contributed to improving our work. We're glad that our response addressed your concerns and appreciate your positive evaluation.

---

> ### Author Response · Authors · 2025-08-04
> **A Gentle Reminder of the Post-rebuttal Discussion**
>
> Dear Reviewer Hewn,
>
> We sincerely thank you for your valuable and constructive comments. We hope our response has adequately addressed your concerns and see this as a valuable opportunity to further improve our work. We would greatly appreciate any additional feedback you might have on our rebuttal :)
>
> Best Regard,
>
> Author(s)

---

### Official Review · Reviewer_9Gaf · 2025-06-30

**Clarity:** 2
**Significance:** 2
**Originality:** 3
**Rating:** 4
**Confidence:** 4

**Summary:**

This paper introduces ALIGNED-IS, a distortion-free statistical watermarking method tailored for autoregressive audio generation models. It tackles the critical retokenization mismatch issue arising from the non-deterministic nature of tokenization and re-tokenization in audio models. The core idea is to cluster similar tokens, based on continuous audio feature similarity, so that during both generation and detection, tokens within the same cluster are treated as equivalent. This cluster-level sampling is paired with a newly proposed Aligned Inverse Sampling strategy to embed watermarks without distorting model output distributions. Extensive experiments demonstrate the method’s competitive detectability, robustness under audio attacks, and minimal impact on audio quality.

**Questions:**

1. Could the authors provide quantitative analysis on how clustering alleviates mismatch? For example, what is the match rate before and after clustering, or the percentage of mismatched tokens still falling into the same cluster?
2. What is the computational overhead of the clustering step (once per model) and aligned inverse sampling (per generation)? A comparison to KGW or γ-reweight would be helpful.
3. Could the authors report detection performance across different numbers of clusters (e.g., h=10, 20, 50) and include a non-clustering baseline?
4. Can the clustering-based approach generalize to other generation models (e.g., multimodal or image) where feature similarity might not correspond to token similarity?

**Ethical Concerns:**

["NO or VERY MINOR ethics concerns only"]

**Final Justification:**

The response addressed my concerns. However, the technique contribution is somewhat limited. Thus, I slightly raise my score.

**Limitations:**

Yes

**Quality:**

3

**Strengths And Weaknesses:**

Strengths

- The method tackles a fundamental limitation, retokenization mismatch, in applying statistical watermarks to audio generation models. The use of token clustering and aligned inverse sampling is both effective and intuitive.

- The authors test on multiple audio models (SpeechGPT, SpiritLM) and datasets, comparing detectability, robustness, and fidelity against state-of-the-art methods.

---

Weaknesses

- Writing quality: Minor writing issues are present (e.g., lines 42–45 are repeated verbatim).

- Insufficient analysis on clustering effectiveness: The paper does not quantify how much the clustering mitigates retokenization mismatch. Metrics like how well tokens match before and after clustering, or how often mismatched tokens end up in the same cluster, could help better support the claims.
- Limited practical advantage: Compared to KGW(δ=1.5), ALIGNED-IS offers similar detectability and robustness, but with extra preprocessing (clustering) and sampling overhead, which may limit practical deployment. Arguably, its only advantage is better fidelity, but this advantage is not particularly significant.
- Generalization and scalability concerns: It remains unclear whether similar clustering properties hold for other modalities (e.g., image models) or more audio models. The paper could benefit from discussion or experiments on extensibility.

---

> ### Author Rebuttal · Authors · 2025-07-29
>
> Thank you for the detailed and thoughtful comments, we're glad that you found our approach to retokenization mismatch both effective and intuitive. Below we address the concerns.
>
> > W1. Writing quality: Minor writing issues are present (e.g., lines 42–45 are repeated verbatim).
>
> **A1.** Thank you for pointing out, we will fix it in our revision.
>
> > W2. Insufficient analysis on clustering effectiveness: The paper does not quantify how much the clustering mitigates retokenization mismatch. Metrics like how well tokens match before and after clustering, or how often mismatched tokens end up in the same cluster, could help better support the claims.
>
> **A2.** Thank you for the suggestion. To quantify the effectiveness of our clustering method in mitigating retokenization mismatch, we report the token mismatch rates before and after applying clustering on SpiritLM across multiple datasets. The mismatch rate is computed by comparing the original response tokens with those obtained after a decode-then-encode retokenization process. As shown in the table, our clustering method significantly reduces the mismatch rate, demonstrating its effectiveness in aligning tokens and preserving fidelity during generation.
>
> | Dataset | Mismatch Rate Before | Mismatch Rate After | Reduction (%) |
> |----------------|----------------------|---------------------|---------------------|
> | mmw_book_report| 0.3749 | 0.2117 | 43.55% |
> | mmw_story | 0.3652 | 0.2174 | 40.47% |
> | mmw_fake_news | 0.4295 | 0.2300 | 46.44% |
> | dolly_cw | 0.3634 | 0.2134 | 41.30% |
> | longform_qa | 0.3757 | 0.2109 | 43.85% |
> | finance_qa | 0.3587 | 0.2133 | 40.54% |
>
> > W3. Limited practical advantage: Compared to KGW(δ=1.5), ALIGNED-IS offers similar detectability and robustness, but with extra preprocessing (clustering) and sampling overhead, which may limit practical deployment. Arguably, its only advantage is better fidelity, but this advantage is not particularly significant.
>
> **A3.** We respectfully disagree with this claim. While ALIGNED-IS introduces a **one-time** clustering overhead, it offers a key practical advantage over KGW(δ=1.5) by better preserving the original model’s distribution. As shown in Figure 2, KGW(δ=1.5) noticeably degrades generation quality (e.g., higher FAD scores), whereas ALIGNED-IS maintains fidelity. This fidelity preservation is critical for service providers who require watermarking without compromising user experience. Furthermore, based on [1], we evaluated additional quality metrics: speaker similarity  and ASR-WER, and observed that ALIGNED-IS consistently outperforms KGW(δ=1.5) in these aspects, reinforcing its practical value beyond detectability and robustness. Baseline is the content generated without watermark.
>
> | Method      | Speaker Similarity ↑ | ASR-WER ↓ |
> |-------------|-----------------------|-----------|
> | Baseline    | 0.7550        | 0.1475    |
> | Aligned-IS  | 0.7548     | 0.1472    |
> | KGW(1.5) | 0.7446   |  0.1617    |
>
> Besides, the one-time cost of the clustering step is 0.3935s for Spiritlm and 0.4150s for Speechgpt. For audio generation, both ALIGNED-IS and KGW have a computational complexity of $O(VL)$, where $V$ denotes the size of the token set and $L$ is the generation length. Empirically, ALIGNED-IS and KGW introduce negligible (<0.01 s/token) computational overhead to the generation process. Thus, they have similar time efficiency.
>
> > W4. Generalization and scalability concerns: It remains unclear whether similar clustering properties hold for other modalities (e.g., image models) or more audio models. The paper could benefit from discussion or experiments on extensibility.
>
> **A4.** Our approach is indeed extensible beyond audio models. For example, autoregressive image generation models such as Emu3 [2] also employ an image-token encoder-decoder structure, analogous to audio tokenization. We conducted a clustering analysis on Emu3 and observed similar retokenization mismatch patterns, which our clustering method effectively mitigates (see the table below). This demonstrates that the underlying principles of our approach generalize to other modalities that rely on discrete token representations, supporting its broader applicability and scalability.
> In this table, we calculated the retokenization mismatch rate with and without clustering method on Emu3 with 500 images from LAION dataset:
>
> | Mismatch Rate Before | Mismatch Rate After | Reduction (%) |
> |-------|------|-----|
> | 0.5131 | 0.2305 | 55.07% |
>
>
>
> > Q1. Could the authors provide quantitative analysis on how clustering alleviates mismatch? For example, what is the match rate before and after clustering, or the percentage of mismatched tokens still falling into the same cluster?
>
> **A.** See **A2**.
>
> > Q2. What is the computational overhead of the clustering step (once per model) and aligned inverse sampling (per generation)? A comparison to KGW or γ-reweight would be helpful.
>
> **A.** The time cost of the clustering step is 0.3935s for Spiritlm and 0.4150s for Speechgpt. The clustering method does not cost a lot because 1) we use the basic k-means clustering and 2) the number of speech tokens is small (500~1000). For audio generation, ALIGNED-IS, KGW, and γ-reweight all have a computational complexity of $O(VL)$, where $V$ is the size of the token set and $L$ is the generation length. Similar to KGW and γ-reweight, our method add negligible (<0.01 s/token) computational overhead to the generation process.
>
> Q3. Could the authors report detection performance across different numbers of clusters (e.g., h=10, 20, 50) and include a non-clustering baseline?
>
> **A.** We conducted an ablation study on the number of clusters for SpiritLM using the Dolly CW dataset, with generation settings identical to the main experiments. The results show that detectability initially increases with more clusters but begins to decline beyond a certain point. This trend arises because SpiritLM has approximately 500 audio tokens, and using too many clusters leads to overly fine partitions that fail to effectively mitigate retokenization errors. We find that $h=20$ strikes a good balance and yields optimal detectability, supporting our choice in the main experiments. For the non-clustering baseline, we assign tokens randomly to h=20 clusters and apply ALIGNED-IS.
>
> |  h  |   TPR@FPR=1%   | Median p-value |
> |:--:|:------:|:--------:|
> | non-cluster | 0.6112 | 0.0059 |
> | 10 | 0.7983 | 0.00047 |
> | 20 | 0.9220 | 2.27e-05 |
> | 40 | 0.9044 | 6.96e-05 |
> | 100 | 0.8586 | 0.00026 |
> | 150 | 0.6879 | 0.0029 |
>
> Q4. Can the clustering-based approach generalize to other generation models (e.g., multimodal or image) where feature similarity might not correspond to token similarity?
>
> **A.** Yes, the clustering-based approach can generalize to other generation models, including multimodal or image models, even when feature similarity does not align with token similarity. In such cases, we can identify mismatched token pairs by comparing the original token sequence with the one obtained after a decode-then-encode roundtrip. Using these mismatches, we construct a graph where nodes represent tokens and edges connect mismatched pairs. The connected components of this graph naturally define clusters that reflect the model’s retokenization behavior, allowing our method to adapt without relying on direct feature similarity.
>
> [1]  O’Reilly et al. Deep Audio Watermarks are Shallow: Limitations of Post-Hoc Watermarking Techniques for Speech. ICLR GenAI workshop.
>
> [2] Wang et al. Emu3: Next-Token Prediction is All You Need, https://arxiv.org/abs/2409.18869.

---

> ### Author Response · Authors · 2025-08-04
> **A Gentle Reminder of the Post-rebuttal Discussion**
>
> Dear Reviewer 9Gaf,
>
> We sincerely thank you for your valuable and constructive comments. We hope our response has adequately addressed your concerns and see this as a valuable opportunity to further improve our work. We would greatly appreciate any additional feedback you might have on our rebuttal :)
>
> Best Regard,
>
> Author(s)

---

### Official Review · Reviewer_g2s4 · 2025-07-03

**Clarity:** 3
**Significance:** 3
**Originality:** 3
**Rating:** 4
**Confidence:** 4

**Summary:**

ALIGNED-IS proposes a watermarking method for AR audio generation models. Its main contribution is achieving distortion-free watermarking despite the retokenization mismatch problem that affects direct token-based methods. To address this, the method clusters similar audio tokens and embeds watermarks at the cluster level.

The main algorithmic contribution is Aligned Inverse Sampling (AIS), a sampling strategy that aligns cluster-level probabilities with fixed intervals for watermark detection, ensuring robustness without altering the model's output distribution.

AIS is proven to be statistically identical (in expectation) to the original model's distribution over clusters. Experiments show that AIS outperforms prior distortion-free methods in detectability and robustness.

**Questions:**

- Since AIS hinges on clustering, can you clarify why clustering is the right abstraction for addressing retokenization mismatch in AR audio models? Were other approaches considered?
- Cluster-level watermarking likely reduces bit capacity (Def 2). Can you quantify this and discuss the impact on latency, detectability, or real-world throughput?
- Can you evaluate AIS under adversarial conditions particularly attacks that manipulate cluster boundaries e.g., pitch shift or adaptive retokenization?

NOTE: I am at the borderline and will evaluate based on the rebuttal.

**Ethical Concerns:**

["NO or VERY MINOR ethics concerns only"]

**Final Justification:**

Overall, I am satisfied with the rebuttal. While the use of clustering is a design choice, the empirical evidence suggests it is a valid one for this problem. However, it still carries inherent limitations, so I am keeping my recommendation at borderline accept.

**Limitations:**

No. The authors do not include any explicit discussion of limitations or potential negative societal impact.

**Paper Formatting Concerns:**

Nothing major

**Quality:**

3

**Strengths And Weaknesses:**

Strengths
- The paper introduces a novel Aligned Inverse Sampling (AIS) thats tailored to the challenges of watermarking AR audio models, addressing the unique retokenization mismatch problem that prior methods overlook.
- The method is theoretically grounded (Theorem 4.1) and empirically validated on audio perturbations and outperforms prior distortion-free baselines.
- Generally, the paper is well-structured and technical components like AIS and clustering are explained with reasonably.
- Its a timely contribution toward trustworthy content attribution in generative systems esp where provenance is important.

Weaknesses
- AIS requires access to token embeddings for clustering, and cant be used in black-box system scenarios.
- Limited to AR models, which narrows applicability
- Cluster-level watermarking reduces bit capacity per token, but the paper does not quantify this tradeoff.
- The paper does not evaluate/acknowledge adversarial attacks, which is important especially for real-time audio, including adversarial stream injection attacks.

---

> ### Author Rebuttal · Authors · 2025-07-29
>
> Thank you for your thoughtful and positive review. We greatly appreciate your recognition of our contributions, both in addressing the retokenization mismatch challenge and in advancing trustworthy content attribution for generative audio systems. Below we address the concerns.
>
> > W1. AIS requires access to token embeddings for clustering, and cant be used in black-box system scenarios.
>
> **A1.** Clustering does not need to be performed online. The service providers run the clustering once, offline, using any model for which they have embedding access, and then publish the deterministic token→cluster lookup table (the only thing the generator/detector needs at run time). At deployment, both watermarking and detection use only token IDs and the precomputed mapping, so no embedding access is required in the black‑box setting. This is identical in spirit to prior watermarking schemes that precompute hash-based green/red lists and only apply light-weight reweighting at inference.
> Even in the extreme case, where the token embeddings are completely inaccessible, we can still identify mismatched token pairs by comparing the original token sequence with the one obtained after a decode-then-encode roundtrip. Using these mismatches, we can construct a graph where nodes represent tokens and edges connect mismatched pairs. The connected components of this graph naturally define clusters that reflect the model’s retokenization behavior, allowing our method to adapt without relying on token embeddings.
>
> > W2. Limited to AR models, which narrows applicability
>
> **A2.** Our goal and evaluation scope is explicitly AR models, which is one of the most powerful categories of speech generation systems. That said, the core idea (cluster the discrete tokens space and watermark at the cluster level) naturally extends to any model that emits a sequence of discrete tokens. For fully non-AR paradigms (e.g., diffusion or masked LM decoding), one can instantiate the same idea over iterative refinement steps (treating each step as a “pseudo time step”) or over latent-code clusters (for VQ-tokenized models). We will make this scope explicit and briefly sketch these adaptations.
>
> > W3. Cluster-level watermarking reduces bit capacity per token, but the paper does not quantify this tradeoff.
>
> **A3.** Thank you for pointing this out. We agree that cluster-level watermarking reduces the bit capacity per token compared to token-level watermarking schemes. This trade-off is an intentional design choice to mitigate retokenization mismatch, which is particularly problematic in auto-regressive audio generation models. While token-level schemes can encode up to 1 bit per token (e.g., by selecting from two disjoint sets per token), cluster-level watermarking limits the number of distinct watermark codes to the number of clusters $h$, leading to roughly $h/|V|$ bit per token (|V|) is the volume of the vocabulary, thus reducing the number of independent bits that can be embedded across a generation.
>
> To make this concrete, we consider the following example. Suppose we generate a sequence of $L=100$ tokens:
> In token-level watermarking, each token can carry 1 bit. If there is no token collision, this will result in a theoretical capacity of 100 bits. In cluster-level watermarking with $h=20$ clusters, each cluster can be assigned a watermark bit, and all tokens within that cluster share the same bit. If the 100 tokens span roughly 20 clusters, the maximum number of independent watermark bits is bounded by 20.
>
> | Method | Bit Capacity per Token | Max Bits in 100 Tokens | Comments |
> | --- | -- | --- | -- |
> | Token-level watermark | \~1.0 bit | 100 bits | High capacity, but prone to retokenization mismatch |
> | Cluster-level (h=20) | ≤0.2 bits | ≤20 bits | Lower capacity, but more robust to retokenization |
>
> Despite the lower bit capacity, our empirical results show that ALIGNED-IS achieves strong detectability while maintaining generation quality, which demonstrates that the reduced capacity is sufficient in practice. We will include this discussion and quantification in the revised version.
>
> > W4. The paper does not evaluate/acknowledge adversarial attacks, which is important especially for real-time audio, including adversarial stream injection attacks.
>
> **A4.** We kindly point the reviewer to Section 5.2, where we provide a comprehensive evaluation of AIS under 13 common attacks in audio watermarks, including common signal processing modifications, popular audio codings, and the denoising attack. Following the authors suggestion, we consider one stream injection attack: The threat model is that the attacker cuts the stream at time t and inserts tokens/frames generated by an unwatermarked model, then resumes. Below, we report the detection results of AIS with different injection strength, where $\epsilon$ is the injection ratio (i.e., injection length / original length). We see that AIS can achieve good detectability under 30% stream injection attack.
>
> SpiritLM, Librispeech
> | $\epsilon$  | TPR@FPR=1% |
> |-:|--:|
> | 0.1  | 0.9036  |
> | 0.2 | 0.8319 |
> | 0.3  | 0.7482 |
>
> SpeechGPT, LongformQA
> | $\epsilon$  | TPR@FPR=1% |
> |--:|---:|
> | 0.1 | 0.9247 |
> | 0.2| 0.8593 |
> | 0.3  | 0.7735 |
>
> > Q1. Since AIS hinges on clustering, can you clarify why clustering is the right abstraction for addressing retokenization mismatch in AR audio models? Were other approaches considered?
>
> **A.** To quantify the effectiveness of our clustering method in mitigating retokenization mismatch, we report the token mismatch rates before and after applying clustering on SpiritLM across multiple datasets. The mismatch rate is computed by comparing the original response tokens with those obtained after a decode-then-encode retokenization process. As shown in the table, our clustering method significantly reduces the mismatch rate, demonstrating its effectiveness in aligning tokens during watermarking generation and detection.
>
> | Dataset | Mismatch Rate Before | Mismatch Rate After | Reduction (%) |
> |---|-----|---|----|
> | mmw_book_report| 0.3749 | 0.2117 | 43.55% |
> | mmw_story | 0.3652 | 0.2174 | 40.47% |
> | mmw_fake_news | 0.4295 | 0.2300 | 46.44% |
> | dolly_cw | 0.3634 | 0.2134 | 41.30% |
> | longform_qa | 0.3757 | 0.2109 | 43.85% |
> | finance_qa | 0.3587 | 0.2133 | 40.54% |
>
> We have also evaluated other popular clustering methods, e.g., (Gaussian mixture models, spectral clustering) and found that they produce similar results. We did not explore methods beyond clustering, as it is the most natural and direct approach to address the retokenization mismatch. Empirically, clustering has proven effective in significantly reducing such mismatches.
>
> > Q2. Cluster-level watermarking likely reduces bit capacity (Def 2). Can you quantify this and discuss the impact on latency, detectability, or real-world throughput?
>
> **A.** The quantification of bit capacity is in **A3**.
>
> **Impact on latency and throughput:** Our method introduces minimal latency overhead, as the clustering is performed once offline on the tokenizer’s vocabulary. At inference time, the only additional cost is mapping each token to its corresponding cluster, which is a simple hash lookup. Thus, real-world throughput is essentially unaffected.
>
> **Impact on detectability:** While cluster-level watermarking reduces the number of independent bits, it significantly improves alignment under decode-then-encode mismatch, which is crucial in modalities like audio or image generation. As shown in our experiments, ALIGNED-IS achieves strong detectability despite reduced capacity, indicating that the trade-off is well-balanced for practical deployment.
>
> > Q3. Can you evaluate AIS under adversarial conditions particularly attacks that manipulate cluster boundaries e.g., pitch shift or adaptive retokenization?
>
> **A.** We have evaluated AIS under a variety of adversarial conditions that can manipulate cluster boundaries. As shown in Tables 4–5 and 19–28, AIS and other baselines are tested against a range of perturbations including Speed Perturbation, EnCodec re-encoding, Gaussian Noise, and Smoothing. These attacks alter temporal or spectral properties of the waveform in ways that can shift token assignments across clusters. In particular, speed perturbation effectively causes pitch and timing shifts, while EnCodec and Gaussian noise introduce quantization and feature distortions, each capable of disrupting the token cluster boundaries. These evaluations demonstrate the robustness of AIS in realistic scenarios where adaptive retokenization effects are present. We also provided one additional pitch shift attack and two low-bitrate-based EnCodec attacks on SpeechGPT and LongformQA datasets to provide further evidence:
>
> SpeechGPT, LongformQA
>
> | Pitch Shift Attack     | TPR@FPR=1% |
> |------------------|-------------|
> | +1 Semitone      | 0.9171     |
> | -1 Semitone      | 0.9215      |
>
> | EnCoder Attack     | TPR@FPR=1% |
> |------------------|-------------|
> | FACodec [1]  (1.6 kbps) | 0.9268 |
> | Speech tokenizer [2]  (4 kbps)| 0.9085 |
>
>
> > L1. The authors do not include any explicit discussion of limitations or potential negative societal impact.
>
> **A.** We kindly point the reviewer to Appendix A, where we discuss the limitations of our work.
>
> Potential negative societal impact: While our method improves the robustness and detectability of watermarks in autoregressive audio generation, it may also raise concerns regarding surveillance and misuse. For instance, persistent watermarking of generated speech, especially when applied without user consent, could be exploited to trace or deanonymize users in voice-based applications. We will put a detailed discussion in our revision.
>
> [1] Ju, Zeqian, et al. "Naturalspeech 3: Zero-shot speech synthesis with factorized codec and diffusion models." arXiv preprint arXiv:2403.03100 (2024).
>
> [2] Zhang, Xin, et al. "Speechtokenizer: Unified speech tokenizer for speech large language models." ICLR 2024.

---

> ### Author Response · Authors · 2025-08-04
> **A Gentle Reminder of the Post-rebuttal Discussion**
>
> Dear Reviewer g2s4,
>
> We sincerely thank you for your valuable and constructive comments. We hope our response has adequately addressed your concerns and see this as a valuable opportunity to further improve our work. We would greatly appreciate any additional feedback you might have on our rebuttal :)
>
> Best Regard,
>
> Author(s)

---

### Decision · Program_Chairs · 2025-09-17

**Decision:**

Accept (poster)

**Comment:**

The paper proposes ALIGNED-IS, a distortion-free watermarking method for autoregressive audio generation.

Strengths:
- Innovative solution to retokenization mismatch using token clustering and aligned inverse sampling.
- Demonstrates strong robustness against adversarial and codec-based attacks.
- Maintains high audio fidelity and speaker consistency across evaluations.

Weaknesses:
- Limited to AR models

I recommend acceptance, as the paper offers a novel and robust watermarking method with strong empirical results and high audio fidelity.